# Conformational changes, excess area, and elasticity of the Piezo protein-membrane nanodome from coarse-grained and atomistic simulations

**Sneha Dixit[1], Frank Noé[2], Thomas R Weikl[1]***

[1]Department of Biomolecular Systems, Max Planck Institut of Colloids and Interfaces, Potsdam, Germany; [2]Department of Mathematics and Computer Science, Freie Universität Berlin, Berlin, Germany

## eLife Assessment

This work represents an **important** contribution to our understanding of how membrane energetics influence protein conformation and function in mechano-sensitive channels. Through extensive molecular dynamics simulations and energetic analysis, the study **convincingly** demonstrates how the channel structure is shaped by a balance of protein and membrane-induced forces, effectively reconciling experimental data from different membrane environments. This work will appeal broadly to researchers and readers with interests in ion channel structure and function, mechanosensation, and membrane biophysics.

**\*For correspondence:**
thomas.weikl@mpikg.mpg.de

**Abstract** The mechanosensitive ion channels Piezo 1 and 2 induce a curved protein-membrane nanodome that flattens with increasing membrane tension $\gamma$. The tension-induced flattening of the nanodome is associated with Piezo activation and driven by the energy $\gamma\Delta A$ where $\Delta A$ is the excess area of the curved nanodome relative to its planar projected area. Based on extensive coarse-grained and atomistic simulations of membrane-embedded Piezo 1 and 2 proteins, we report here an excess area $\Delta A$ for the Piezo protein-membrane nanodome of about 40 $nm^2$ in tensionless membranes, and a half-maximal reduction of $\Delta A$ at tension values of about 3–4 mN/m, which is within the range of experimentally determined values for the half-maximal activation of Piezo 1. In line with recent experimental investigations of Piezo proteins in cell membranes and membrane vesicles, the membrane-embedded Piezo proteins adopt conformations in our simulations that are significantly less curved than the protein conformation in the detergent micelles of cryo-EM structures. An elasticity analysis of the nanodome shapes and protein conformations obtained from our simulations leads to an elastic model for Piezo activation that distinguishes the different energy components of the protein and the membrane in the tension-induced flattening of the nanodome. According to this model, the Piezo proteins resist flattening with a force constant of about 60 pN/nm.

## Introduction

The transmembrane (TM) proteins Piezo 1 and 2 are mechanosensitive ion channels (*Coste et al., 2010*; *Coste et al., 2012*) that mediate numerous physiological processes in mammals, including touch sensation and blood pressure control (*Wu, 2017*). Cryo-electron microscopy (cryo-EM) structures of Piezo 1 (*Guo and MacKinnon, 2017*; *Saotome et al., 2018*; *Zhao et al., 2018*) and Piezo 2

(*Wang et al., 2019*) in detergent micelles revealed three identical monomeric arms with 38 TM helices that spiral out from the central ion channel, which is lined by the innermost TM helices of the arms. The TM domains of the helices in the three arms do not lie in a plane, which led to the suggestions that the Piezo protein curves the cell membrane into a nanodome, and that this protein-membrane nanodome flattens when external forces induce a tension $\gamma$ in the membrane, leading to channel opening (*Guo and MacKinnon, 2017*). The flattening of the nanodome is driven by the energy $\gamma \Delta A$, where $\Delta A$ is the excess area of the nanodome due to its curved shape, compared to the projected area of the nanodome in the plane of the surrounding membrane (*Guo and MacKinnon, 2017*; *Haselwandter and MacKinnon, 2018*). Cryo-electron tomography images of reconstituted membrane vesicles with embedded Piezo 1 proteins indeed show that the proteins induce membrane curvature (*Guo and MacKinnon, 2017*), but elastic modeling of vesicle shapes with varying diameters indicates that this induced curvature is on average about four times smaller than the curvature of the proteins in the detergent micelles of the high-resolution cryo-EM structures (*Haselwandter et al., 2022a*; *Haselwandter et al., 2022b*). High-resolution fluorescence imaging of Piezo 1 in its native cell membrane environment confirms an expansion in the inactivated state from flattening of the arms compared to structural models in detergent micelles and indicates further flattening upon activation (*Mulhall et al., 2023*). A cryo-EM structure of flattened Piezo 1 with widened ion channel has been obtained by embedding Piezo 1 proteins in small membrane vesicles with a diameter of 20 nm in an outside-out orientation in which the vesicle curvature opposes the intrinsic Piezo curvature (*Yang et al., 2022*). In patch-clamp experiments with Piezo 1 and Piezo 2, an opening of the ion channel induced by membrane tension has been observed for Piezo 1, but not for Piezo 2, which appears to indicate that Piezo 2 activation requires other factors of its cellular context that are not present in the experiments, besides membrane tension (*Moroni et al., 2018*).

Molecular dynamics (MD) simulations have the potential to complement recent experimental insights on the structure of Piezo proteins in their native membrane environment by providing high-resolution, dynamic information of membrane-embedded Piezo proteins under different membrane tensions (*Botello-Smith et al., 2019*; *Chong et al., 2021*; *Buyan et al., 2020*; *Lin et al., 2022*; *Jiang et al., 2021*; *De Vecchis et al., 2021*). In atomistic simulation trajectories with a length up to 100 ns at membrane tensions between 14.2 mN/m and 67.8 mN/m starting from a tensionless Piezo 1 protein-membrane nanodome with a simulation-box area of 31.4 × 31.4 nm$^2$, *De Vecchis et al., 2021* observed a flattening of the nanodome, and a significant expansion of the channel volume at the largest tension of 67.8 mN/m. In atomistic simulations of truncated Piezo 1 with TM helices 17–38 in a tensionless membrane, a flattening of the protein-membrane nanodome leading to channel opening has been induced by a small simulation-box area of about 480–485 nm$^2$ that aims to mimic a crowding of Piezo 1 proteins (*Jiang et al., 2021*). The membrane flattening in these simulations is a consequence of the standard periodic boundary conditions of the simulation box, which lead to an on-average vanishing slope of the membrane at the box boundaries.

In this article, we report results from coarse-grained simulations of the Piezo 1 and Piezo 2 protein-membrane nanodome with a length up to 8 μs at membrane tensions between 0 and 20.8 mN/m, starting from tensionless protein-membrane nanodomes with a simulation-box area of 50 × 50 nm$^2$. To corroborate these coarse-grained simulation results, we performed additional atomistic simulations of the Piezo 2 protein-membrane nanodome with a length up to 300 ns and an initial, tensionless simulation-box area of 33 × 33 nm$^2$ at membrane tensions between 0 and 18 mN/m. A main focus in our analysis is on determining the excess area $\Delta A$ of the protein-membrane nanodomes as a function of the membrane tension $\gamma$. In both our coarse-grained simulations with the Martini 2.2 force field (*de Jong et al., 2013*) and atomistic simulations with the CHARMM36 force field (*Huang and MacKerell, 2013*; *Klauda et al., 2010*), we obtain an excess area $\Delta A$ for the protein-membrane nanodome of about 40 nm$^2$ in the tensionless state, a reduction to values of about 5 nm$^2$ and below at tension values larger than 10 mN/m at which the nanodome is nearly completely flattened, and a half-maximal response of $\Delta A$ at tension values of about 3–4 mN/m, which is within the range of experimentally determined values for the half-maximal activation of Piezo 1 (*Lewis and Grandl, 2015*; *Cox et al., 2016*). In agreement with experimental observations for membrane-embedded Piezo 1 (*Haselwandter et al., 2022a*; *Haselwandter et al., 2022b*; *Mulhall et al., 2023*), the excess area $\Delta A$ as well as the height and curvature of the tensionless protein-membrane nanodomes is significantly reduced compared to corresponding values for the cryo-EM structure of the proteins in detergent

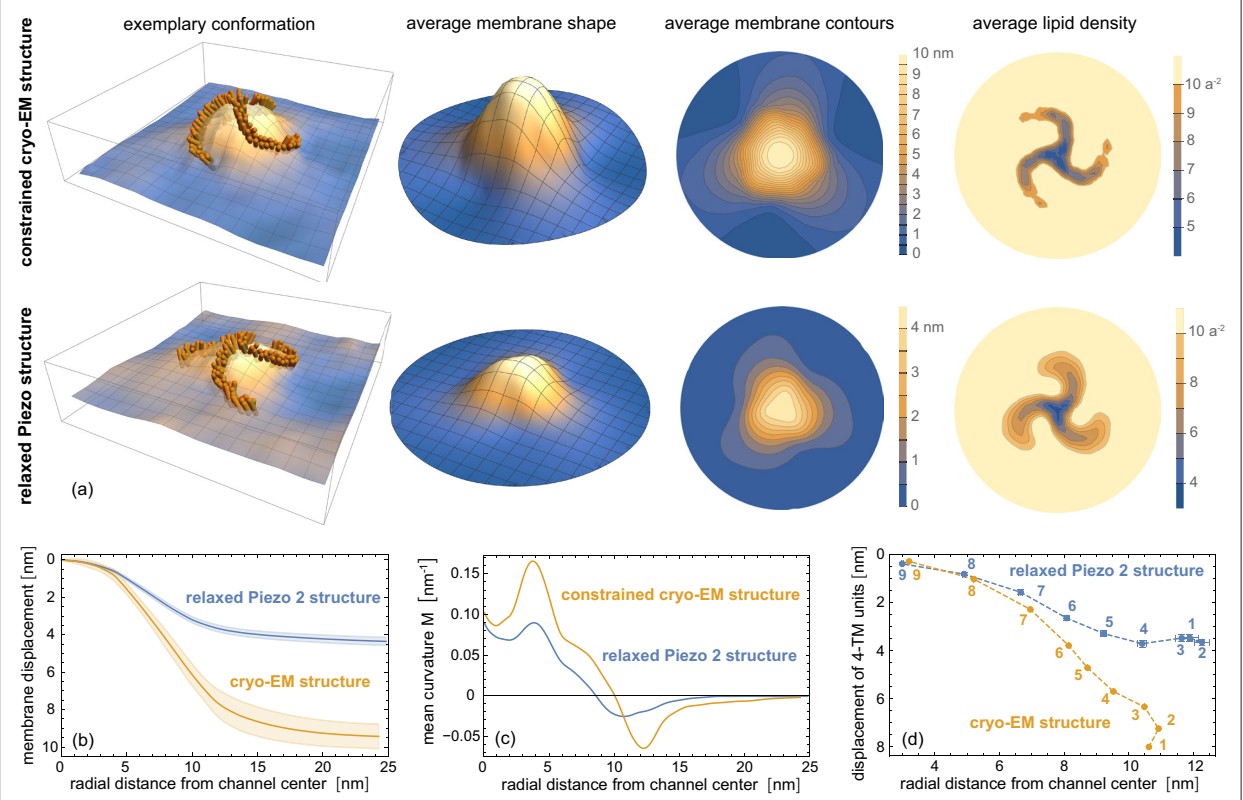

**Figure 1.** Coarse-grained Piezo 2 nanodome in a tensionless membrane. (**a**) Representative conformations and average nanodome shape, average nanodome contours, and average lipid density within quadratic membrane patches with side length $a \simeq 2$ nm for the constrained Piezo 2 cryo-EM structure and the relaxed Piezo 2 structure in a coarse-grained tensionless membrane. The relaxed nanodome shape is an average over conformations from the last microseconds of 10 independent simulation runs with a length of 8 µs. The nanodome shape for the constrained Piezo 2 cryo-EM structure is an average over the last microseconds of three independent simulation runs with a length of 4 µs for a harmonic force constant of 1000 kJ mol$^{-1}$ nm$^{-2}$ per bead. (**b**) Radial membrane profiles of the average nanodome shapes. The shaded areas represent the error of the mean obtained from the shape profiles of the independent runs. (**c**) Mean curvature $M$ along the shape profiles calculated from first and second derivatives at radial distance $r$, which are obtained from local quadratic fits of profile segments $r \pm 2$ nm. (**d**) Average displacement of the center of mass (COM) of 4 TM-helix units versus average radial distance. The 4 TM-helix units are numbered as in **Figure 7**. Only protein residues in the TM sections of the 4-TM units are included in the calculation of COMs (see Materials and methods).

micelles. At high membrane tensions, we observe a widening of the Piezo 1 ion channel as in the flattened cryo-EM structure of Piezo 1 in small membrane vesicles (**Yang et al., 2022**). The Piezo 2 ion channel, in contrast, does not respond to membrane tension in our simulations, which is in line with patch-clamp experiments in which Piezo 1, but not Piezo 2, was shown to be activated by membrane tension alone (**Moroni et al., 2018**). In addition to $\Delta A$, we determine the bending energy of the lipid membrane in the nanodomes, which allows us to construct an elasticity model that identifies the energetic contributions of the protein and the membrane in the response of the Piezo protein-membrane nanodome to membrane tension.

## Results
### Piezo 2 protein structure and nanodome shape in tensionless membranes

Our simulations of the Piezo 2 protein-membrane nanodome are based on the cryo-EM structure of the Piezo 2 trimer (PDB ID 6KG7) (**Wang et al., 2019**), which includes all 38 TM helices of the Piezo 2 monomers. In these simulations, the Piezo 2 trimer is embedded in an asymmetric membrane that mimics essential aspects of the lipid composition of cell membranes (see Materials and methods). We find that the relaxed simulation structure of Piezo 2 in tensionless membranes is significantly flattened

compared to the cryo-EM structure of the Piezo 2 trimer in detergent micelles (see *Figure 1*). To compare these structures, we have performed coarse-grained simulations of membrane-embedded Piezo 2 in which the protein structure is constrained to the cryo-EM structure, in addition to simulations without such constraints in which the protein structure relaxes in the tensionless membrane within the 8 µs of the simulation trajectories. The two representative conformations in *Figure 1a* compare the protein-membrane nanodome at the end of simulation trajectories with and without constraints on the protein structure. In these conformations, the TM helices of Piezo 2 are indicated as rods, and the shape of the membrane midplane is shown as a continuous surface determined from the positions of the lipid head beads in the two leaflets of the membranes (see Materials and methods). The relaxed simulation structure of Piezo 2 is clearly less curved than the constrained cryo-EM structure. In addition to these different curvatures of the protein-membrane nanodome, the membrane shape in both representative conformations of *Figure 1a* exhibits undulations from thermally excited shape fluctuations.

To quantify the protein-induced curvature and excess area of the nanodome, and to eliminate the additional curvature and excess area of thermal membrane shape fluctuations present also in protein-free membranes, we average the continuous membrane-midplane shapes of individual conformations over the same time intervals of all trajectories. The average membrane shapes, membrane contours, and lipid densities in *Figure 1a* are averaged over the conformations in the last microseconds of all 10 simulation trajectories of length 8 µs with the relaxed Piezo 2 protein, and over the last microseconds of the three simulation trajectories with the constrained protein (see Materials and methods). Because of the translational and rotational diffusion of the Piezo proteins along the simulation trajectories, the averaging of conformations requires an alignment of the protein (see Materials and methods). The average membrane shapes and lipid densities in *Figure 1a* therefore have a circular projected area with a diameter that corresponds to the average $x$ and $y$ dimensions of the simulation boxes of the conformations, because only these 'inscribed' circular membrane segments are overlying in all rotationally aligned conformations. From the simulation trajectories in which the protein structure is constrained to the cryo-EM structure of Piezo 2 in detergent micelles, we obtain a nanodome midplane shape with a height of 10 nm, which is significantly larger than the height of about 4 nm for the nanodome shape with the relaxed Piezo 2 protein. The mean curvature profiles of the nanodome shapes in *Figure 1c*, which we calculate from the radially averaged nanodome profiles of *Figure 1b*, exhibit maxima at a radial distance of about 4 nm from the channel. In these mean curvature profiles, the maximal mean curvature of the nanodome with constrained protein structure is about two times larger than the maximal mean curvature of the nanodome with the relaxed protein. The mean curvature of the nanodome with the relaxed Piezo 2 protein drops to 0 for radial distances larger than about 17 nm from the channel center, which indicates a catenoidal shape of the non-planar surrounding membrane with a principal curvature in the radial direction of the shape profile in *Figure 1b* that is oppositely equal to the principal curvature in the perpendicular, tangential direction (*Haselwandter and MacKinnon, 2018*).

## Tension-induced flattening of the protein-membrane nanodome

With increasing membrane tension, the nanodome shape and coarse-grained simulation structure of the membrane-embedded Piezo 2 protein flattens further (see *Figure 2*). The average height of the nanodome decreases from 4.4 nm in tensionless membranes to about 3.5 nm, 2.9 nm, and 2.1 nm at the intermediate membrane tensions $\gamma$ of 1.4 mN/m, 2.8 mN/m, and 5.5 mN/m of our simulations, and to 1.6 nm and 0.9 nm at the largest tensions of 10.8 mN/m and 20.8 nM/m (see radial nanodome profiles in *Figure 2d*). In atomistic simulations of membrane-embedded Piezo 2, we observe a comparable tension-induced flattening of the Piezo 2 protein-membrane nanodome, and a comparable height of the nanodome in tensionless membranes, which corroborates our coarse-grained simulation results (see *Figure 3*).

To obtain values of the excess area $\Delta A$ that are not limited by the finite simulation time, in particular for the atomistic simulations, we divide all simulation trajectories into five equal intervals and determine the nanodome shape in each interval by averaging over the conformations of all independent simulation runs in this interval. In *Figure 4a-c*, we plot the excess area $\Delta A$ of the nanodome shape in the last time intervals versus the inverse time $t^{-1}$ of the interval midpoints, for our coarse-grained simulations of membrane-embedded Piezo 1 and Piezo 2 proteins and our atomistic simulations of the

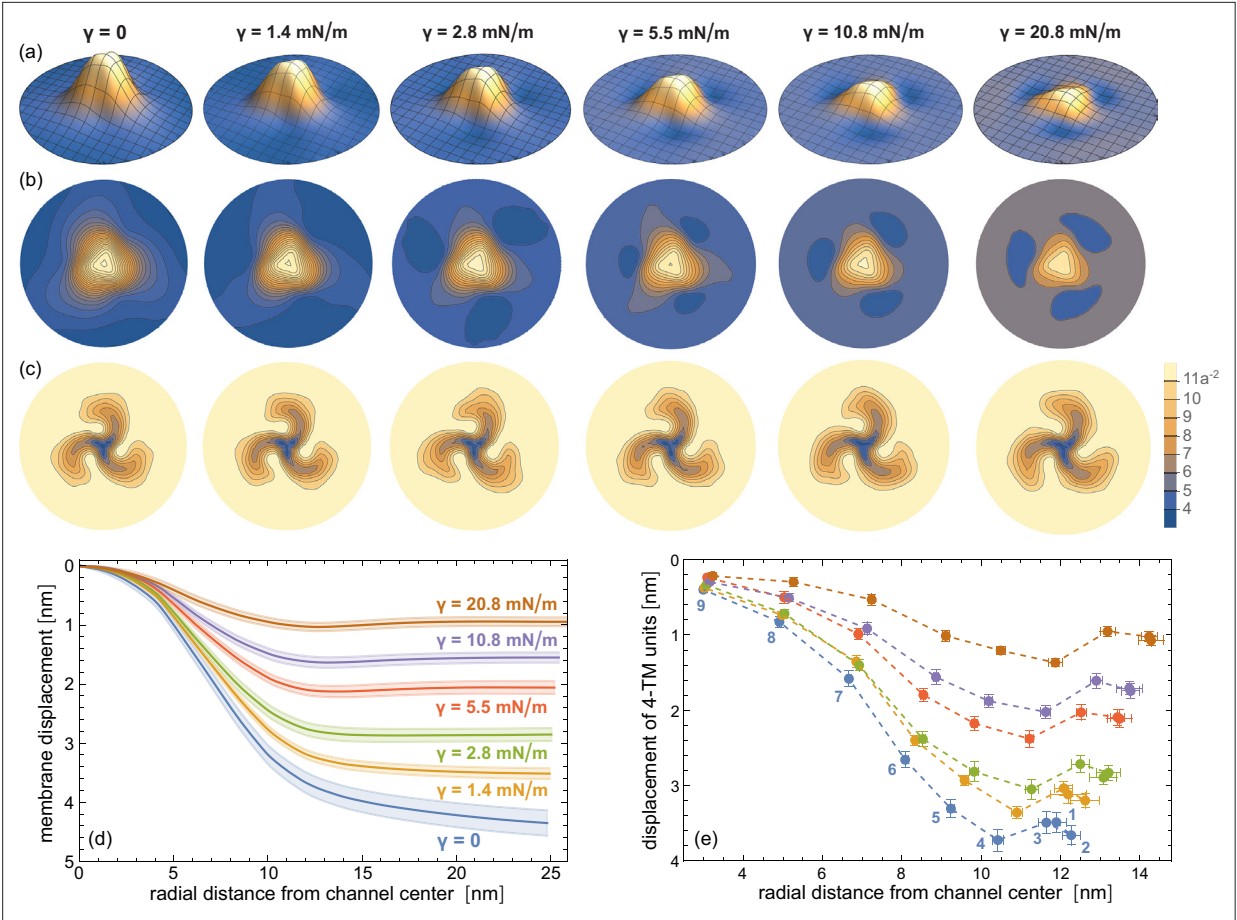

**Figure 2.** Tension-induced flattening of the coarse-grained Piezo 2 protein-membrane nanodome. (**a**) Average nanodome shapes, (**b**) average nanodome contours, and (**c**) average lipid density within quadratic membrane patches with sidelength $a \simeq 2$ nm at membrane tensions $\gamma$ from 0 to 20.8 mN/m. The nanodome shapes and lipid densities are averages over conformations from the last microseconds of 10 independent simulation runs with lengths of 8 μs for $\gamma = 0$ and with lengths of 4 μs for $\gamma > 0$. (**d**) Radial membrane profiles of the average nanodome shapes in (**a**). The shaded areas represent the error of the mean obtained from the shape profiles of the independent runs. (**e**) Average displacement of the center of mass (COM) of 4 TM-helix units versus average radial distance. The 4 TM-helix units are numbered as in *Figure 7*. Only protein residues in the TM sections of the 4-TM units are included in the calculation of COMs.

membrane-embedded Piezo 2 protein. The excess area $\Delta A$ here was calculated as the area difference between the curved nanodome shape and the planar projection of the nanodome (see Materials and methods). As starting points of our coarse-grained simulations of membrane-embedded Piezo 1, we devised a structure of full-length Piezo 1 by adding the unresolved 12 N-terminal TM helices to the Piezo 1 cryo-EM structure with PDB ID 6B3R (*Guo and MacKinnon, 2017*) *via* homology modeling (see Materials and methods). In all simulation systems, we observe a decrease of $\Delta A$ with increasing simulation time, and therefore extrapolate the $\Delta A$ values to $t^{-1} = 0$ by linear fitting (dashed lines and shaded prediction error bands). In *Figure 3e*, the extrapolated $\Delta A$ values are plotted versus the membrane tension $\gamma$ in the simulations. In all simulation systems, we obtain a maximal excess area $\Delta A$ for the protein-membrane nanodome of about 40 nm² in tensionless membrane, and half-maximal $\Delta A$ values at tensions of about 3–4 mN/m, which is within the range of experimentally determined values for the half-maximal activation of Piezo 1 (*Lewis and Grandl, 2015*; *Cox et al., 2016*). The nanodome excess area $\Delta A$ of about 40 nm² in tensionless membranes is significantly smaller than the excess area $\Delta A = 165 \pm 1$ nm² of the average membrane nanodome shape with constrained cryo-EM structure of Piezo 2 shown in *Figure 1(a)*.

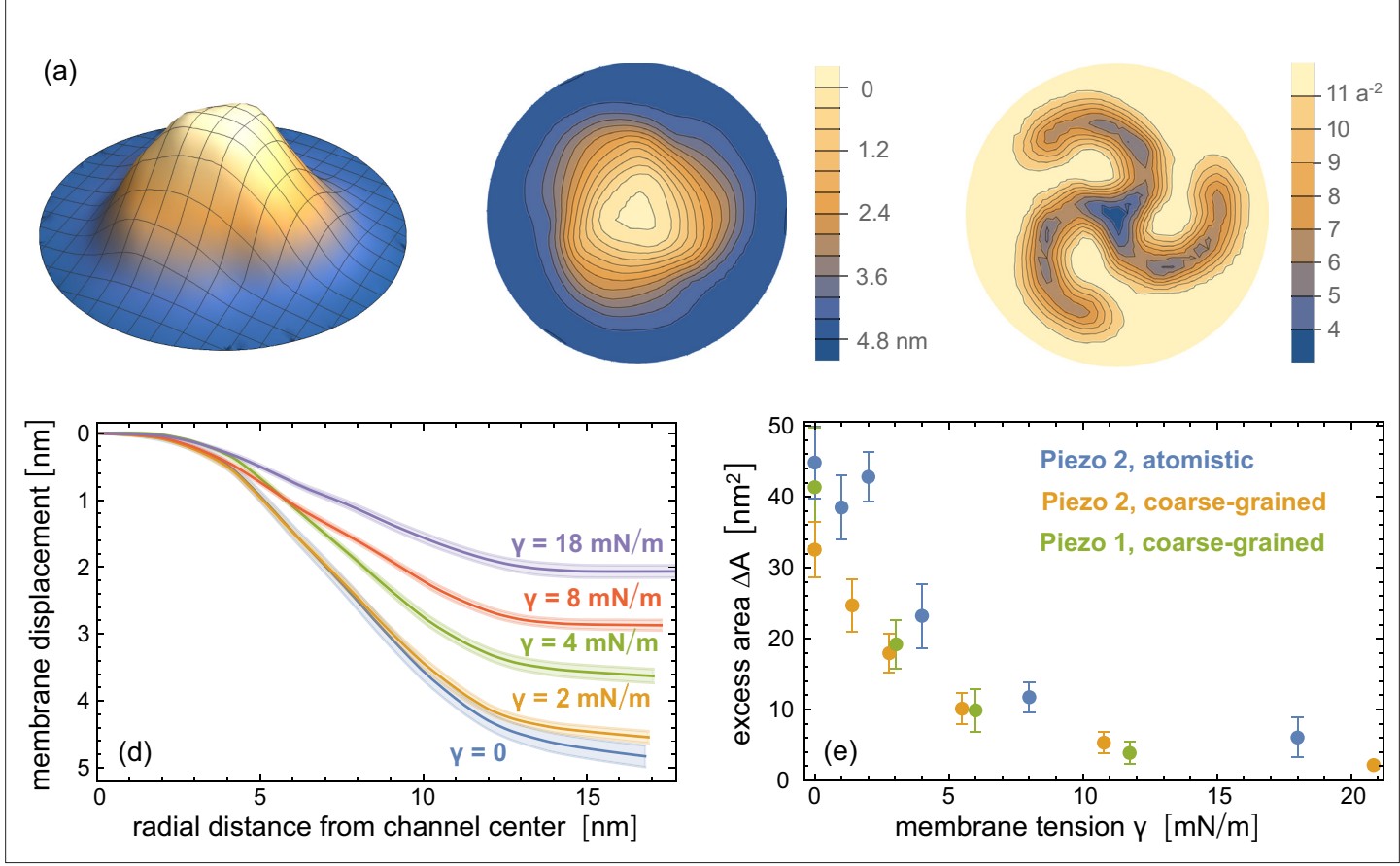

**Figure 3.** Flattening of the Piezo 2 protein-membrane nanodome in atomistic simulations. (**a**) Average nanodome shape, (**b**) average nanodome contours, and (**c**) average lipid density in a tensionless membrane. The nanodome shapes and lipid densities are averages over conformations from the last 50 ns of 5 independent atomistic simulation runs with lengths of 300 ns. (**d**) Radial membrane profiles of the nanodome shapes at membrane tensions $\gamma$ from 0 to 18 mN/m obtained from averaging over conformations from the last 50 ns of five independent atomistic simulation runs. (**e**) Excess area $\Delta A$ of the protein-membrane nanodome versus membrane tension $\gamma$ from coarse-grained simulations of membrane-embedded Piezo 1 and Piezo 2 and atomistic simulations with Piezo 2. The values and errors of excess area $\Delta A$ are obtained from the extrapolations to long timescales shown in *Figure 4*.

## Elasticity model of the Piezo protein-membrane nanodome

We now aim to construct a model for the elastic energies of the Piezo protein and the membrane in the tension-induced nanodome flattening observed in our simulations. The nanodome height in tensionless membranes and the tension-induced flattening of the nanodome are a consequence of opposing protein and membrane energies. The energy of the membrane is the sum $E_m = E_b + \gamma \Delta A$ of the membrane bending energy $E_b$ and the tension-associated energy $\gamma \Delta A$, and 'prefers' a planar state in which this energy tends to zero. The energy $E_p$ of the Piezo protein conformation, in contrast, prefers a curved nanodome state. The nanodome height is determined by the sum of these energies (*Guo and MacKinnon, 2017*; *Haselwandter and MacKinnon, 2018*), that is by the total energy

$$E_{\text{tot}} = E_p + E_m = E_p + E_b + \gamma \Delta A \tag{1}$$

To describe the tension-dependent Piezo protein conformation, we first note that the vertical displacement $z_4$ of the fourth 4-TM unit of the Piezo 2 arms relative to the channel center is larger than the vertical displacements $z_i$ of the other 4-TM units in *Figure 2e*. We therefore use the vertical displacement $z_4$ of 4-TM unit 4 to describe the vertical height of the Piezo 2 TM domain, and the shift $\Delta z(\gamma) = z_4(0) - z_4(\gamma)$ of this displacement in comparison to the tensionless nanodome as parameter for the tension-induced flattening of Piezo 2. *Figure 5a* illustrates that the Piezo 2 height change $\Delta z$

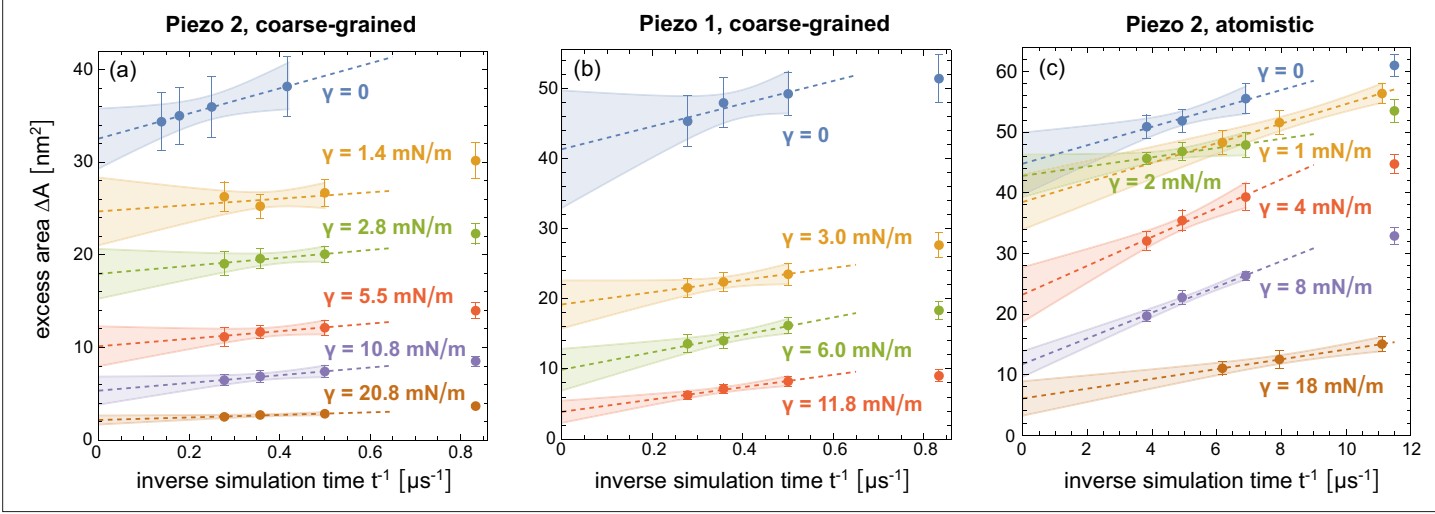

**Figure 4.** Extrapolation of the excess area $\Delta A$ of the protein-membrane nanodome in coarse-grained simulations with (**a**) Piezo 2 and (**b**) Piezo 1 and (**c**) in atomistic simulations with Piezo 2 to long timescales. In these extrapolations, the trajectories lengths are divided into five equal time intervals, and the excess area $\Delta A$ in each time interval is calculated for the nanodome shape obtained from averaging over conformations of all independent simulation runs in this time interval. The $\Delta A$ values are plotted against the inverse of the centers of the time intervals, and the values of the last three intervals are linearly fitted with the function LinearModelFit of Mathematica 13. The errors of the data points represent the error of the mean of $\Delta A$ values obtained for the individual simulation runs, and the shaded error region of the linear fits represents prediction bands with confidence level $0.5$. The $\Delta A$ values with error shown in Figure 4 are determined as the extrapolated values with prediction band error at $t^{-1} = 0$.

The online version of this article includes the following figure supplement(s) for figure 4:

**Figure supplement 1.** Extrapolation of the membrane bending energy $E_b$ of the nanodome in coarse-grained simulations with (**a**) Piezo 2 and (**b**) Piezo 1 to long timescales, akin to the extrapolations of $\Delta A$ in *Figure 4*.

in our coarse-grained simulations is proportional to the membrane tension $\gamma$ for tension values up to 6 mN/m.

At the equilibrated Piezo 2 height obtained at a given tension value, the vertical forces resulting from the protein elasticity and from the membrane elasticity are oppositely equal. If this was not the case, the Piezo 2 height would further increase or decrease until such a force balance is achieved at this tension value. We can therefore calculate the vertical force exerted by the flattening-resisting protein from the opposing force exerted by the membrane. At a given membrane tension $\gamma$, the vertical force resulting from the membrane energy $E_m$ is

$$F = \left.\frac{\mathrm{d}E_m}{\mathrm{d}\Delta z}\right|_{\gamma=const.} = \frac{\mathrm{d}E_b}{\mathrm{d}\Delta z} + \gamma\frac{\mathrm{d}\Delta A}{\mathrm{d}\Delta z} \tag{2}$$

This force is the sum of two terms, and calculating these terms requires determining the membrane bending energy $E_b$ and the excess area $\Delta A$ of the nanodome as functions of the Piezo height change $\Delta z$. In *Figure 5b*, we plot the $\Delta A$ values obtained from the coarse-grained simulations of membrane-embedded Piezo 2 at different tensions versus the $\Delta z$ values of the protein at these tensions, which indicates a linear relation within the error margins obtained from the simulations. From the linear fit in *Figure 5(b)*, we obtain $\mathrm{d}\Delta A/\mathrm{d}\Delta z = -14.6 \pm 1.8$ nm.

For calculating the membrane bending energy of the averaged nanodome shapes, we use a discretized version of the bending energy and include the lipid densities shown in *Figure 2(c)* in order to limit the bending energy calculations to the lipid membrane of the nanodome (see Materials and methods). To avoid limitations due to the finite simulation times, we extrapolate the bending energy values obtained for the average nanodome shapes in 5 intervals of the simulation trajectories to long timescales akin to the temporal extrapolations of the excess area $\Delta A$ (see *Figure 4—figure supplement 1*). The resulting, extrapolated bending energy values $E_b$ of the lipid membrane in the Piezo 2 protein-membrane nanodome decrease linearly with the Piezo 2 height change $\Delta z$ (see *Figure 5c*). From linear fitting, we obtain $\mathrm{d}\Delta E_b/\mathrm{d}\Delta z = -0.82 \pm 0.17\ \kappa/$nm where $\kappa$ is the bending rigidity of the lipid membrane. For the coarse-grained membrane of the lipid composition used in our simulations

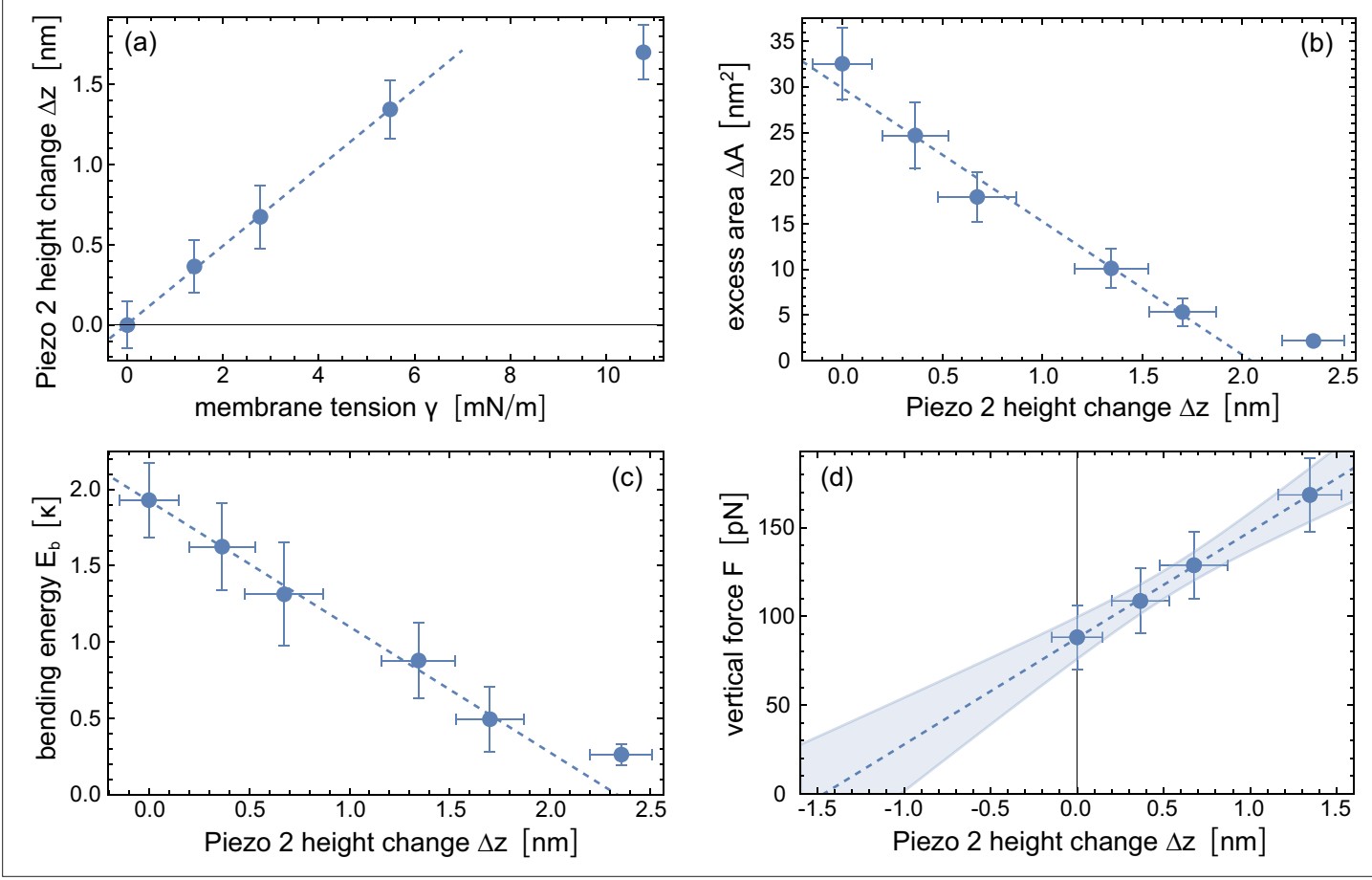

**Figure 5.** Elasticity modeling of the Piezo 2 protein based on coarse-grained simulation data. (a) Piezo 2 height change $\Delta z$ versus membrane tension $\gamma$ in our coarse-grained simulations of membrane-embedded Piezo 2. The protein height change is defined as $\Delta z(\gamma) = z_4(0) - z_4(\gamma)$ where $z_4$ is the vertical displacement of the fourth 4-TM unit of the Piezo arms relative to the channel center (see *Figure 2e*). (b) Excess area $\Delta A$ of the nanodome and (c) bending energy $E_b$ of the lipid membrane in the nanodome versus Piezo height change $\Delta z$. The values and errors of $\Delta A$ and $E_b$ are obtained from extrapolating simulation results at the tension values $\gamma = 0, 1.4, 2.8, 5.5, 10.8$, and $20.8$ mN/m to long timescales (see *Figure 4*, *Figure 4—figure supplement 1*). (d) Vertical force $F$ determined from *Equation 2* and the slopes of the fitted lines in (b) and (c) at the $\Delta z$ values obtained from simulations at the tension values $\gamma = 0, 1.4, 2.8$, and $5.5$ mN/m. The vertical force $F$ is the absolute value of the opposing and in equilibrium equal forces exerted by the Piezo protein and by the membrane at a given tension value (see text). The dashed lines result from linear fitting of data points for $\gamma \leq 6$ mN/m in (a), of data points for $\Delta z < 2$ nm in (b) and (c), and of all data points in (d) with the function LinearModelFit of Mathematica 13. The shaded error region of the linear fit in (d) represents prediction bands with confidence level $0.5$.

The online version of this article includes the following figure supplement(s) for figure 5:

**Figure supplement 1.** Elasticity modeling of the Piezo 1 protein based on coarse-grained simulation data.

with main component POPC, we estimate a bending rigidity value of about $\kappa = 25$ k$_B$T based on rigidity values obtained from coarse-grained simulations with the same force field Martini 2.2 for membranes composed of POPC and of various lipid mixtures (*Fowler et al., 2016*; *Pöhnl et al., 2023*; *Hossein et al., 2024*).

From *Equation 2* and the derivatives d$\Delta A$/d$\Delta z$ and d$\Delta E_b$/d$\Delta z$ determined above *via* linear modeling, we obtain the four data points in *Figure 5d* for the four $\Delta z$ values at the membrane tensions $\gamma = 0$, $1.4$, $2.8$, and $5.5$ mN/m, which indicate a linear increase of the force $F$ with the Piezo height change $\Delta z$. From linear fitting, we obtain the force $F(\Delta z = 0) = 88 \pm 15$ pN for the tensionless membrane and the force constant d$F$/d$\Delta z = 60 \pm 20$ pN/nm. This linear force relation is consistent with a harmonic-spring model for the elasticity of the Piezo protein. Extrapolation to zero force in *Figure 5d* indicates a force-free state of Piezo 2 for $\Delta z = -1.5 \pm 0.6$ nm, which is – within the statistical accuracy – in accordance with the difference of $-2.0 \pm 0.2$ nm between the vertical displacement $z_4(\gamma = 0) = 3.7 \pm 0.2$ nm of the membrane-embedded Piezo 2 protein in tensionless membranes and the vertical displacement

$z_4 = 5.7$ nm in the cryo-EM structure of Piezo 2 in detergent micelles (see *Figure 1d*). Our force modeling thus is consistent with the Piezo cryo-EM structure in detergent micelles as force-free conformation of the protein. In this modeling, the term $F(\Delta z = 0)$ depends linearly on the estimated bending rigidity value, whereas the force constant $dF/d\Delta z$ is independent of $\kappa$. For a smaller bending rigidity value of $\kappa = 20$ $k_BT$ instead of 25 $k_BT$, we obtain $F(\Delta z = 0) = 70 \pm 12$ pN and a force-free state of Piezo 2 for $\Delta z = -1.2 \pm 0.5$ nm. Our modeling assumes that any spontaneous curvature from asymmetries in the lipid composition is small compared to the curvature of the nanodome and, thus, negligible, which is plausible for the rather slight lipid asymmetry of our simulated membranes (see Materials and methods).

From our coarse-grained simulation data of membrane-embedded Piezo 1, we obtain a linear force relation with force $F(\Delta z = 0) = 113 \pm 28$ pN in the tensionless state and a force constant $dF/d\Delta z = 63 \pm 30$ pN/nm for $\kappa = 25$, which agrees with our results for Piezo 2 within errors (see *Figure 5—figure supplement 1*). The larger errors in the force relation for Piezo 1 compared to Piezo 2 are also a consequence of fewer tension values considered in our coarse-grained simulations of membrane-embedded Piezo 1 and, thus, fewer data points in the modeling.

## Response of the Piezo 1 and 2 ion channel to large membrane tensions

Curved and flattened cryo-EM structures obtained for Piezo 1 embedded in small membrane vesicles with outside-in and outside-out protein orientation indicate a widening of the Piezo 1 ion channel in

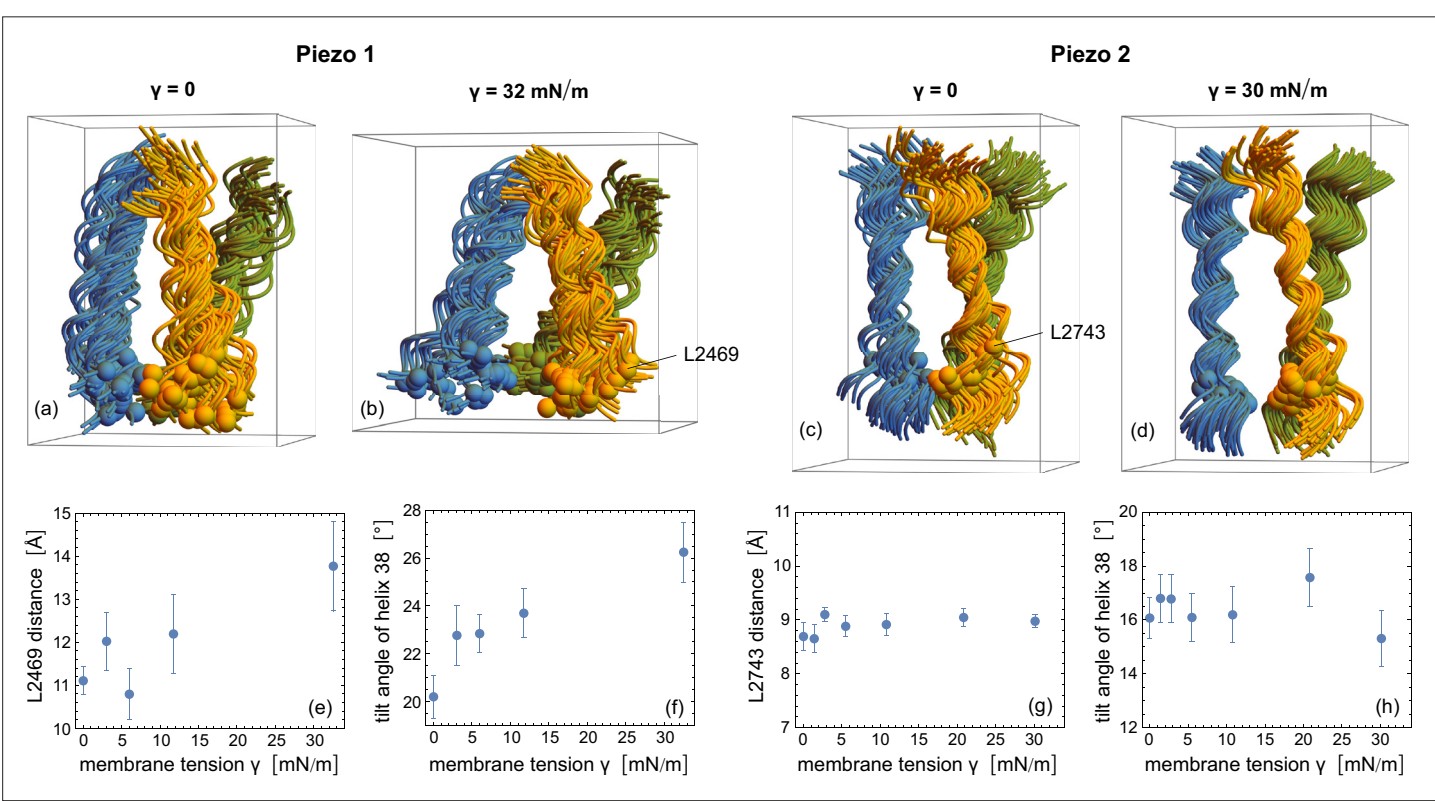

**Figure 6.** Response of the Piezo 1 and Piezo 2 channels to membrane tension. (**a,b**) Simulation conformations of the three TM helices 38 that line the ion channel from our coarse-grained simulations of membrane-embedded Piezo 1 at vanishing membrane tension $\gamma = 0$ and at the largest simulated tension $\gamma = 32$ mN/m. (**c,d**) Simulation conformations of the TM helices 38 from our coarse-grained simulations of membrane-embedded Piezo 2 at $\gamma = 0$ and the largest simulated tension $\gamma = 30$ mN/m. The 50 aligned simulation conformations in (**a**) to (**d**) are taken from the last microseconds of the 10 simulation trajectories at intervals of 0.25 µs along each trajectory (5 conformations per trajectory), with TM helices 38 depicted as spline curves of backbone beads in Mathematica 13. (**e**) Average distance between the backbone beads of the residues L2469 that form the outermost constriction site of the ion channel in Piezo 1 (*Yang et al., 2022*) versus membrane tension $\gamma$ of our simulations. The L2469 backbone beads of the three TM helices 38 are shown as beads in (**a**) and (**b**). (**f**) Average tilt angle of the Piezo 1 helices 38 relative to the vertical $z$ direction of the simulation conformations in (**a,b**) versus membrane tension $\gamma$ of our simulations. (**g**) Average distance between the backbone atoms of the residues L2473 that form the outermost constriction site of the ion channel in Piezo 2 (*Yang et al., 2022*) versus membrane tension $\gamma$. (**h**) Average tilt angle of the Piezo 2 helices 38 versus membrane tension $\gamma$. Error bars in (**e**) to (**h**) represent errors of the mean calculated from average values obtained for the 10 trajectories.

response to flattening (*Yang et al., 2022*). In the outside-in orientation, the extracellular side of Piezo 1 points towards the inside of the vesicles with mean diameter of 20 nm, which results in a highly curved Piezo structure. In the outside-out orientation, the extracellular side of Piezo 1 points towards the vesicle outside, and the Piezo protein structure is nearly completely flattened because the vesicle curvature opposes the intrinsic Piezo curvature in this orientation. In the flattened Piezo 1 structure, the distance between the residues L2469 of the three TM helices 38 that line the ion channel is increased by 5 Å compared to the curved Piezo 1 structure, which indicates a widening of the outer-most constriction site of the Piezo 1 ion channel formed by these residues (*Yang et al., 2022*).

In our coarse-grained simulations of membrane-embedded Piezo 1, we observe a comparable widening of the ion channel lined by the three TM helices 38 in response to tension-induced flattening of the Piezo 1 protein. At the largest simulated membrane tension $\gamma = 32$ mN/m, the Piezo 1 protein is nearly completely flattened, and the distance between the channel-constricting residues L2469 of the three TM helices 38 is increased by about $3 \pm 1$ Å compared to the Piezo 1 channel in our tensionless membranes (see *Figure 6*). This increased distance between the residues L2469 at large membrane tensions is associated with an increased tilt angle of the TM helices 38 relative to the vertical $z$ direction of our simulation conformations. In our coarse-grained simulations of membrane-embedded Piezo 2, in contrast, we do not observe a response of the ion channel to tension (see *Figure 6*), apparently in line with patch-clamp experiments of Piezo 1 and Piezo 2, in which a tension-induced opening of the ion channel has only been observed for Piezo 1, but not for Piezo 2 (*Moroni et al., 2018*).

## Discussion

Based on coarse-grained and atomistic simulations of membrane-embedded Piezo proteins, we have investigated the shape and excess area $\Delta A$ of the protein-membrane nanodome as well as the conformation of the Piezo proteins in tensionless membranes and for a physiologically relevant range of membrane tensions. Our coarse-grained simulations of membrane-embedded full-length Piezo 1 and 2 proteins allowed us to consider relatively large segments of about $50 \times 50$ nm² on simulation timescales up to 8 μs. To assess and corroborate the validity of these coarse-grained simulations, we performed atomistic simulations of membrane-embedded full-length Piezo 2 in smaller membrane segments of about $33 \times 33$ nm² on simulation timescales up to 300 ns. To quantify the protein-induced curvature and excess area of the nanodome, and to eliminate the additional curvature and excess area of thermal membrane shape fluctuations, we have determined average nanodome shapes from the simulation conformations (see *Figure 1*). We have determined the excess area $\Delta A$ from extrapolation of $\Delta A$ values for average nanodome shapes in different time intervals of the trajectories to reduce sampling errors from the finite length of the simulation trajectories (see *Figure 4*).

As a main result, we obtained an excess area $\Delta A$ of the Piezo protein-membrane nanodome of about 40 nm² in tensionless membranes from both coarse-grained and atomistic simulations. This excess area of the relaxed protein-membrane nanodome in tensionless membranes is significantly smaller than the excess area $\Delta A = 165 \pm 1$ nm² obtained from our simulations with the constrained cryo-EM structure of Piezo 2, which in turn is comparable in magnitude to the estimate $\Delta A = 250$ nm² from an analysis of the Piezo 2 dimensions in the cryo-EM structure (*Wang et al., 2019*). The excess area $\Delta A$ is reduced to values of about 5 nm² at tension values larger than 10 mN/m, at which the nanodome is nearly completely flattened in our coarse-grained and atomistic simulations, and is half-maximal at tension values of about 3–4 mN/m, which are within the range of experimentally determined values for the half-maximal activation of Piezo 1 (*Lewis and Grandl, 2015*; *Cox et al., 2016*). Based on our results, the change of the nanodome excess area $\Delta A$ required for Piezo activation thus can be estimated to be about 20 nm², which is significantly smaller than previous estimates based on cryo-EM structures, but still large compared to the area difference between the open and closed state of the ion channel and, thus, in line with the suggested flattening-induced Piezo activation (*Guo and MacKinnon, 2017*).

In addition, our elasticity analysis of the nanodome shapes allowed us to construct an elastic model for Piezo activation that distinguishes the different energy components in the tension-induced flattening of the protein-membrane nanodome. According to this model, the Piezo protein elasticity can be approximately described as a harmonic spring, with a force-free conformation that agrees with the curved protein conformation in the cryo-EM structures within the modeling accuracy. The elastic energy of the membrane, which opposes the protein elasticity, is the sum of the membrane bending

energy $E_b$ and the tension-associated energy term $\gamma\Delta A$ that drives the tension-induced flattening. In tensionless membranes, the interplay of the protein elastic energy $E_p$ with the membrane bending energy $E_b$ leads to a flattening of the protein relative to cryo-EM structure that is associated with a force of roughly 100 pN acting on the protein (see *Figure 5d*). With further tension-induced flattening, the force on the protein increases by about 60 pN per nanometer in reduced protein height. With atomic force microscopy (AFM), *Lin et al., 2019* measured the force-induced height change of Piezo1 protein-membrane nanodomes adsorbed to substrates. In this system, the force-induced height change is not only affected by the elastic energy of the protein and the membrane, but also by the adhesion energy of the membrane to the substrate, which led to the spreading of membrane vesicles with embedded Piezo 1 on the substrate in the generation of the substrate-adsorbed membranes, and may also be affected by the water pocket between the nanodome and the substrate. The smaller force response of about 7 pN per nanometer in reduced nanodome height measured in the AFM experiments, therefore, cannot be compared to the force response of the protein obtained from our modeling of simulation data for the tension-induced flattening of the Piezo protein-membrane nanodome.

Besides the excess area $\Delta A$, our elastic model for Piezo activation is based on the bending energy $E_b$ of the membrane nanodome, which we determined from extrapolations of $E_b$ values for average nanodome shapes in different time intervals of the trajectories (see *Figure 4—figure supplement 1*), akin to the extrapolations of $\Delta A$ values. Compared to $\Delta A$, the bending energies $E_b$ of non-planar membrane shapes are more sensitive to membrane shape undulations because of the scale invariance of $E_b$. This scale invariance implies that the bending energy for small perturbations of the planar state is identical to the bending energy of large perturbations with the same shape, and membrane shape undulations can lead to many such small perturbations. However, the membrane shapes obtained from averaging our coarse-grained simulation conformations are rather smooth and do not exhibit undulations (see *Figures 1 and 2*). To corroborate our bending energy calculations for these averaged three-dimensional nanodome shapes, we note that essentially identical bending energies can be obtained from the highly smoothened mean curvatures $M$ of the two-dimensional membrane profiles. For the mean curvature profile $M$ of the relaxed and tensionless Piezo 2 membrane nanodome in *Figure 1c*, we obtain a bending energy of about 1.7, which agrees within numerical accuracy with the bending energy value $E_b = 1.93 \pm 0.25\,\kappa$ obtained from extrapolation of $E_b$ values for average three-dimensional nanodome shapes (see *Figure 4—figure supplement 1*). At membrane tensions $\gamma = 1.4$, 2.8, 5.5, and 10.8 mN/m, we obtain the bending energies of about 1.4, 1.2, 0.8, and 0.5 $\kappa$, respectively, from two-dimensional mean-curvature profiles $M$ smoothened as in *Figure 1c*, which all agree within numerical accuracy with the bending energy values in *Figure 5c* obtained from three-dimensional shapes. In general, bending energies calculated from two-dimensional nanodome profiles can be expected to be slightly lower than the bending energies determined for the three-dimensional average shapes, because these shapes have the threefold symmetry of the Piezo proteins, but only approximate rotational symmetry (see *Figure 2*).

In tensionless membranes, we obtain a nearly catenoidal shape of the membrane surrounding the Piezo protein nanodome in our coarse-grained simulations, in agreement with calculations for radially symmetric nanodome shapes (*Haselwandter and MacKinnon, 2018*). The mean curvature of the non-planar catenoidal shape is zero because of oppositely equal principal curvatures. We obtain a nearly zero mean curvature for radial distances larger than about 17 nm from the channel center of the relaxed Piezo 2 protein structure (see *Figure 1c*), which indicates that the non-planar membrane profile (see *Figure 1b*) is catenoidal. For tension values equal to or larger than 2.8 mN/m, in contrast, the membrane profile obtained from our coarse-grained simulations is planar for radial distances larger than about 17 nm from the channel center (see *Figure 2d*). This planar membrane profile around the Piezo nanodome results from the interplay of the membrane bending energy and the energy $\gamma\Delta A$ associated with the membrane tension $\gamma$. A characteristic length scale of this interplay is the crossover length $\sqrt{\kappa/\gamma}$. The membrane tension dominates the elastic energy of the membrane on length scales larger than this crossover length. For a membrane tension $\gamma = 1$ mN/m and a bending rigidity $\kappa$ of 25, the crossover length is 10 nm. For $\gamma = 4$ mN/m, the crossover length is reduced to 5 nm. A dominance of membrane tension leads to short-range exponential decay of membrane perturbations out of the planar membrane state, in contrast to the long-range logarithmic decay of the catenoid in tensionless membranes. The results of our coarse-grained simulations thus are independent of the

finite membrane size in the simulations, at least for tension values equal to or larger than 2.8 mN/m. For $\gamma = 0$, the on-average zero membrane slope imposed at the periodic boundaries of the simulation in principle induces deviations from the catenoidal shape. However, these deviations appear to be small for the simulation-box area of about $50 \times 50$ nm$^2$ in our coarse-grained simulations and are not reflected in the mean curvature profile of *Figure 1c* with nearly zero mean curvatures from about 17 nm to the largest radial distances of about 25 nm compatible with the simulation box.

The simulation box area of about $33 \times 33$ nm$^2$ in our atomistic simulations is larger than the simulation box area in previous atomistic simulations (see Introduction), but clearly smaller than the corresponding area of about $50 \times 50$ nm$^2$ in our coarse-grained simulations. In addition, the trajectory lengths up to 300 ns of the computationally significantly more demanding atomistic simulations are clearly shorter than the trajectory lengths between 4 and 8 µs of our coarse-grained simulations. These different membrane areas and trajectory lengths constitute two caveats in the comparison of atomistic and coarse-grained simulation results. To address the smaller size of the atomistic simulations, we first note that the excess area $\Delta A$ of the tensionless nanodome is dominated by the membrane deformations in the center of the nanodome where the protein is located. In the case of the tensionless average shapes obtained from our coarse-grained simulations, limiting the excess area calculation up to radial distances 12 nm from the channel center where the outermost 4-TM helix units are located (see *Figure 1d*) leads to values that are only about 10% smaller than the values for the entire average shape with maximal radial distance of about 25 nm. Limiting the excess area calculation to radial distances of 16.5 nm, which correspond to the maximal radial distance of the atomistic simulations, leads to values that are only about 3% smaller. The smaller membrane area of our atomistic simulations thus appears to be sufficient to capture the main deformations of the nanodome in tensionless membranes. At membrane tensions of 2 mN/m or larger, the membrane around the Piezo protein is already rather flat and contributes only marginally to the overall excess area $\Delta A$ of the nanodome (see profiles in *Figures 2 and 3*). However, for the membrane tensions of 2, 4, and 8 mN/m, the $\Delta A$ values obtained from the extrapolation of atomistic simulation results tend to be somewhat larger than the corresponding $\Delta A$ values obtained in the coarse-grained simulation systems. These differences in the $\Delta A$ values may result from remaining roughnesses in the averaged atomistic conformations (see *Figure 3a and b*), from a larger resistance of the atomistic Piezo 2 protein to tension-induced flattening, or from inaccuracies in the extrapolation of $\Delta A$ values for the atomistic system. The atomistic simulation trajectories start from a membrane shape obtained in coarse-grained simulations after 200 ns of relaxation (see Materials and methods), which is converted to atomistic resolution. Within these 200 ns of relaxation in coarse-grained simulations, the excess area of 165 nm$^2$ for the initial nanodome with cryo-EM Piezo 2 structure as starting structure is reduced by about half. In the five equal time windows of the 300 ns atomistic trajectories for $\gamma = 0$, the averaged shapes have the excess areas $\Delta A = 68 \pm 2, 61 \pm 2, 56 \pm 2, 52 \pm 2$, and $51 \pm 2$ nm$^2$. In *Figure 4c*, the $\Delta A$ values of the last four time windows are plotted as a function of the inverse simulation time, that is of the inverse center of the time windows. From linear extrapolation of these four data points, we obtain the value $\Delta A = 45 \pm 5$ nm$^2$ for $\gamma = 0$. For $\gamma = 4$ and 8 mN/m, the linear extrapolation curves are steeper because the simulation starts from the same structure as the tensionless simulations, which leads to larger uncertainties and potential overestimates in the extrapolation. For the largest membrane tension $\gamma = 18$ mN/m, the excess area is already reduced to about 20 nm$^2$ in the first two time windows of the simulations, followed by smaller reduction in the remaining three time windows with data points included in *Figure 4c*. Overall, the excess area $\Delta A$ clearly decreases in all atomistic simulations relative to the initial structure obtained after short relaxation of 200 ns in coarse-grained simulations. The initial relaxation in coarse-grained simulations facilitates the relaxation towards equilibrium in the atomistic simulations.

# Materials and methods
## Coarse-grained simulations
### Protein modeling
Our Piezo 2 simulation system is based on the cryo-EM structure of the full-length Piezo 2 trimer (PDB ID 6KG7; *Wang et al., 2019*), which includes all 38 TM helices of the Piezo 2 monomers. We added shorter loops of less than 25 residues not resolved in this cryo-EM structure with the software

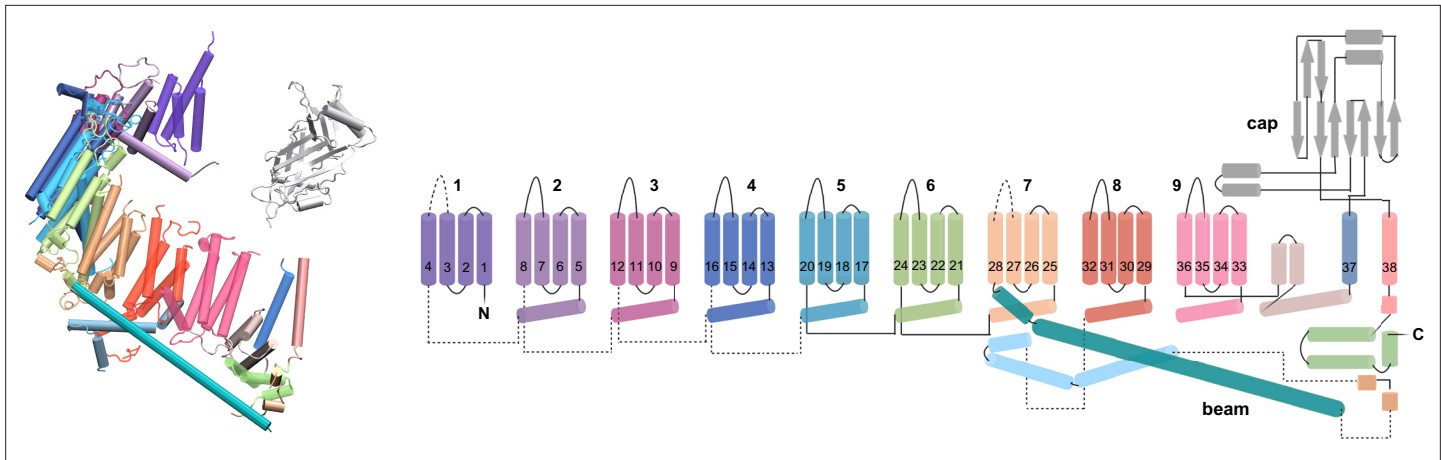

**Figure 7.** Structure and topology of the Piezo 2 monomer (*Wang et al., 2019*). The 38 TM helices of the monomer are numbered from the N- to the C-terminus of the protein chain. The TM helices 1–36 are arranged in 9 TM units with 4 helices each. The ion channel of the Piezo 2 trimer is lined by the three TM helices 38 of the three monomers. Loops not resolved in the cryo-EM structure of the Piezo 2 trimer structure in detergent micelles are indicated by dashed lines.

MODELLER v10.2 (*Sali and Blundell, 1993*; *Webb and Sali, 2016*). The remaining unresolved longer loops not included in our Piezo 2 simulation system are indicated by dashed lines in *Figure 7*. In cryo-EM structures of Piezo 1 (*Guo and MacKinnon, 2017*; *Saotome et al., 2018*; *Zhao et al., 2018*), several of the outer, N-terminal TM helices are not resolved. We therefore used the cryo-EM structure of full-length Piezo2 as template structure to add the unresolved 12 N-terminal TM helices to the Piezo 1 cryo-EM structure with PDB ID 6B3R (*Guo and MacKinnon, 2017*) *via* homology modeling with MODELLER v10.2. Shorter loops of less than 25 residues not resolved in the resulting structure were added as for Piezo 2. To avoid artifacts from missing long protein loops, the distances between the terminal residues of these missing loops were constrained to the distances in the cryo-EM structures using a harmonic potential with spring constant 10 kJ mol$^{-1}$ nm$^{-2}$ in the coarse-grained simulations.

## Force field

We used the Martini 2.2 force field, which exhibits a general 4–1 mapping of non-hydrogen atoms into coarse-grained simulation beads (*de Jong et al., 2013*; *Periole and Marrink, 2013*; *Marrink et al., 2007*; *Monticelli et al., 2008*), in all coarse-grained simulations performed with the software GROMACS 2018.3 (*Abraham et al., 2015*). To allow for tension-induced conformational changes of Piezo 2, we employed the standard protein secondary structure constraints of Martini 2.2 with secondary structures identified with DSSP (*Kabsch and Sander, 1983*), but did not constrain the protein tertiary structure by an additional elastic network (*Periole and Marrink, 2013*). Instead, we relied on the capabilities of the Martini coarse-grained force field for modeling membrane systems with TM helix assemblies (*Sharma and Juffer, 2013*; *Chavent et al., 2014*; *Majumder and Straub, 2021*).

## Membrane embedding

To set up the protein-membrane nanodome of the coarse-grained Piezo 2 simulation system, we first embedded a single monomer of the Piezo 2 trimer into a membrane because the transmembrane region of a Piezo monomer is not strongly curved and can be placed reasonably well in a planar membrane. To this end, we converted the Piezo 2 monomer structure into the coarse-grained Martini

**Table 1.** Lipid percentages of membrane.

|  | POPC | POPE | CHOL | POPS | PIP2 | DPSM |
|---|---|---|---|---|---|---|
| inner leaflet | 50 | 20 | 20 | 5 | 5 | - |
| outer leaflet | 50 | 20 | 20 | - | - | 5 |

2.2 representation (*de Jong et al., 2013*; *Marrink et al., 2007*; *Monticelli et al., 2008*) using the martinize script and generated a coarse-grained planar, asymmetric membrane of area 50 nm × 50 nm with the composition indicated in *Table 1* using the INSert membrANE (INSANE) software script (*Wassenaar et al., 2015*). As in previous simulation studies of membrane-embedded Piezo proteins (*Buyan et al., 2020*; *Chong et al., 2021*; *De Vecchis et al., 2021*; *Jiang et al., 2021*), we chose an asymmetric membrane composition that mimics the cell membrane composition with negatively charged lipids in the inner leaflet.

We then oriented and centered the transmembrane domain of the Piezo 2 monomer along the $x - y$ plane of the membrane and solvated the system with coarse-grained water beads in GROMACS 2018.3 (*Abraham et al., 2015*). To equilibrate the membrane around the Piezo 2 monomer, we constrained the protein beads using harmonic potentials with force constant 10,000 kJ mol$^{-1}$ nm$^{-2}$, performed an energy minimization with 5000 steps of the steepest descent algorithm, and ran a subsequent MD simulation with a length of 50 ns in the NPT ensemble. In this simulation, the pressure was kept at 1 bar using the Berendsen barostat (*Berendsen et al., 1984*) and the temperature was maintained at 310 K using a velocity-rescale thermostat (*Bussi et al., 2007*) with individual coupling

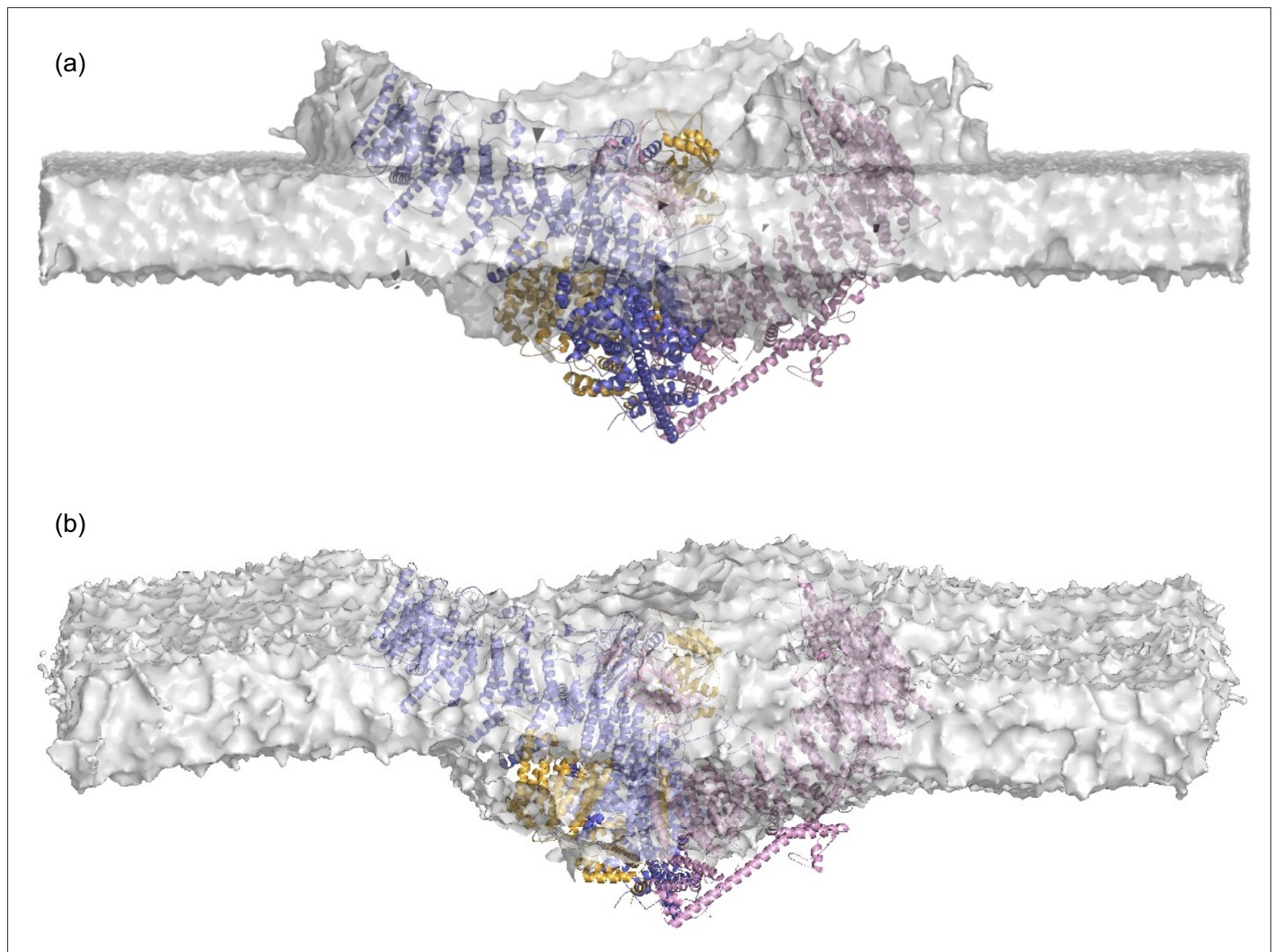

**Figure 8.** Embedding the Piezo 2 protein into a membrane. (**a**) Piezo 2 trimer with 'lipid envelop' placed into a planar and asymmetric membrane of area 50 nm × 50 nm. The lipid envelope was created by insertion of a Piezo 2 monomer into a planar lipid membrane and subsequent superposition of the resulting lipid-enveloped monomers on the Piezo 2 cryo-EM structure (see Materials and methods). (**b**) Piezo 2 trimer embedded in a continuous membrane nanodome after minimization and equilibration simulations starting from (**a**).

of protein, membrane, and solvent. The distance cutoff for the non-bonded interactions was 1.2 nm in this simulation, and bonds were constrained with the LINCS algorithm (*Hess et al., 1997*).

To embed the Piezo 2 trimer, we first generated a Piezo 2 monomer with 'lipid envelope' by selecting all lipid molecules of the membrane-embedded protein monomer that contain at least one bead with distance smaller than 3 nm from a protein backbone bead. We then superimposed this lipid-enveloped monomer along each of the three monomers of the Piezo 2 cryo-EM structure with Pymol 2.5.0 (*Schrödinger, 2015*) and deleted overlapping lipids from the three envelops. We then placed the resulting lipid-enveloped Piezo 2 trimer into a coarse-grained, planar, and asymmetric membrane of area 50 nm × 50 nm with the center of mass of the Piezo 2 ion channel about 4 nm below the membrane midplane (see *Figure 8a*). In this placement, the Piezo 2 trimer remains oriented as in the cryo-EM structure, with the ion channel roughly oriented along the z-axis and, thus, perpendicular to the $x - y$ plane of the membrane. To merge the curved lipid envelope of the Piezo 2 trimer and the planar membrane into a membrane nanodome, we first added water beads and sodium and chloride ions to neutralize the overall system charge at a physiological salt concentration of 150 mM with GROMACS. The system was then energy minimized in 10,000 steps of steepest descent and equilibrated in a 10 ns MD simulation in the NVT ensemble with an integration time step of 10 fs and a subsequent 500 ns MD simulation in the NPT ensemble with an integration time step of 20 fs. In these minimization and equilibrium steps, the protein beads were constrained by a harmonic potential with force constant of 1000 kJ mol$^{-1}$ nm$^{-2}$ to maintain the protein shape. The temperature in the equilibration simulations was kept at 310 K by a velocity-rescale thermostat, and the pressure in the NPT simulations was kept at 1 bar with the Berendsen barostat. During these equilibration simulations, a continuous membrane nanodome is formed (see *Figure 8b*). To embed the Piezo 1 trimer, we took advantage of the structural similarity to Piezo 2 and replaced the membrane-embedded Piezo 2 trimer by the modeled full-length Piezo 1 trimer after structural alignment in Pymol.

## Relaxation

After these membrane embedding and lipid equilibration steps, we slowly released the position restraints on the protein atoms by reducing the force constant to 1000, 100, 10, and 1 kJ mol$^{-1}$ nm$^{-2}$ in four MD simulation steps of 10 ns each in the NPT ensemble with an integration time step of 20 fs. In these simulations, protein, membrane, and solvent beads were coupled independently to an external bath to keep the temperature at 310 K using a velocity-rescale thermostat (*Bussi et al., 2007*). The lateral and normal pressure was maintained at 1 bar using semi-isotropic pressure coupling and the Berendsen barostat (*Berendsen et al., 1984*) with a collision frequency of 5 ps. We implemented bond constraints using the LINCS algorithm (*Hess et al., 1997*) and employed a cut-off of 1.2 nm for the non-bonded interactions. As the last step, we removed the position restraints on protein beads and ran 10 independent trajectories of length 200 ns to generate 10 different input structures for the production simulation runs.

## Coarse-grained production simulations

In the production simulation runs, we switched to the Parrinello-Rahman barostat (*Parrinello and Rahman, 1981*) with a coupling constant of 20 ps to achieve more flexibility for box shape modulations with semi-isotropic pressure coupling (*Braun et al., 2019*). All other simulation parameters were identical to the parameters of the final equilibration simulations. To perform simulations at different membrane tensions, we kept the pressure in the normal $z$ direction at $P_z = 1$ bar and applied a series of pressure values in the lateral $x$ and $y$ directions of the membrane plane. These pressure values were $P_x = P_y = 1, 0.5, 0, -1, -3, -7,$ and $-11$ bar in our coarse-grained Piezo 2 simulations and $P_x = P_y = 1, 0, -1, -3$ and $-11$ bar in our coarse-grained Piezo 1 simulations. The pressure difference in the lateral and normal directions leads to the membrane tension (*Yefimov et al., 2008*)

$$\gamma = L_z \left( P_z - (P_x + P_y)/2 \right) \tag{3}$$

where $L_z$ is the average box height during the simulations. At each lateral pressure, we generated 10 independent production trajectories starting from the same initial conformation obtained after relaxation with a length of 4 μs each. For Piezo 2, we extended the 10 simulation trajectories at zero membrane tension to 8 μs to explore the role of the long-range shape fluctuations in tensionless membranes in the equilibration.

## Atomistic simulations

### System setup

To set up our atomistic simulations of membrane-embedded Piezo 2, we used a coarse-grained Piezo 2 conformation obtained after 200 ns of final equilibration as starting point. The coarse-grained Piezo 2 trimer and surrounding membrane nanodome of this conformation were converted to atomistic resolution with the CHARMM-GUI server (*Wassenaar et al., 2014*; *Jo et al., 2008*; *Brooks et al., 2009*; *Lee et al., 2016*). In this conversion, the phosphoinositol biphosphate (PIP2) of the coarse-grained Martini model was mapped to phosphoinositol (4,5) biphosphate in the atomistic model. We reduced the membrane area to about $33 \times 33$ nm$^2$ to decrease the total number of atoms in the system and removed atomic clashes of lipids and amino acids by energy minimization with 5000 steps of the steepest descent algorithm in GROMACS. We then solvated and neutralized the system at a total salt concentration of 150 mM of KCl by adding an appropriate number of potassium and chloride ions. In our atomistic simulations, we used the CHARMM36 force field (*Huang and MacKerell, 2013*; *Klauda et al., 2010*) with the TIP3P water model (*Jorgensen et al., 1983*).

### System equilibration

We performed the equilibration and production simulations with the Amber20 software (*Case, 2020*) because of the efficient use of GPUs with this software (*Salomon-Ferrer et al., 2013*). To this end, we first generated Amber suitable input coordinates using the ParmEd software tool (*Shirts et al., 2017*) and energy-minimized the system with 10,000 steps of the steepest descent algorithm and subsequent 10,000 steps of the conjugate gradient algorithm. We then heated the system from temperature 0 K to 310 K in three simulation steps of 10 ns in the NVT ensemble using an integration time step of 1 fs and a Langevin thermostat with a collision frequency of 1 ps$^{-1}$. In these three heating simulations, the positions of the protein backbone atoms were restrained by harmonic potentials with a force constant of 10 kcal mol$^{-1}$ Å$^{-2}$. We subsequently released the harmonic restraints in 5 simulation steps of 10 ns in the NPT ensemble, decreasing the force constant by a factor of 2 in each of the steps. In these equilibration simulations with an integration time step of 2 fs, the temperature was maintained at 310 K by a Langevin thermostat with a collision frequency of 1 ps$^{-1}$, the pressure in normal and lateral directions was kept at 1 bar by a Berendsen barostat with a semi-isotropic pressure coupling and a pressure relaxation time of 2 ps, a cutoff length of 10 Å was used for non-bonded interactions, long-range electrostatic interactions were calculated with the Particle Mesh Ewald (PME) method (*Darden, 1993*; *Essmann et al., 1995*), and all bonds involving hydrogen atoms were constrained using the SHAKE algorithm (*Ryckaert et al., 1977*).

### Production simulations

In the production simulations starting from the equilibrated system conformation, we used hydrogen mass repartitioning (*Hopkins et al., 2015*) to increase the integration timestep to 4 fs for computational efficiency. Our production simulations consist of five independent trajectories at each of the membrane tension values 0, 1, 2, 4, 8, and 18 mN/M. In our simulations with the Amber20 software, the membrane tension was implemented by subjecting a constant surface tension at the two interfaces of the lipid bilayer in the $x - y$ plane, using the Berendsen barostat with a reduced pressure relaxation time of 1 ps to ensure rapid pressure regulation under constant surface tension conditions. As in the equilibration simulations, the temperature was maintained at 310 K by a Langevin thermostat with a collision frequency of 1 ps$^{-1}$, bonds containing hydrogen atoms were constrained using the SHAKE algorithm, a cutoff length of 10 Å was used for non-bonded interactions, and long-range electrostatic interactions were calculated with the PME method. At each of the tension values 0, 2, 4, and 8 mN/M, we generated five independent trajectories with a trajectory length of 300 ns starting from the same initial structure. At the tension values 1 and 18 mN/M, we generated five trajectories with a length of 180 ns each.

## Analysis of trajectories

### Membrane shape of simulation conformations

To determine the membrane midplane shape of a coarse-grained simulation conformation, we first translated the conformation so that the center of mass of the ion channel formed by the three helices

38 is located at the center of the simulation box. We next discretized the $x - y$ plane of the simulation box into a square lattice with 25 × 25 equally sized squares (bins), separated the lipid head beads (head beads of POPC, POPE, POPS, POP2, and DPSM) into upper and lower membrane leaflet, and allocated these head beads to bins based on the $x - y$ coordinates of the beads. In each bin with at least two lipid head beads in the upper and in the lower leaflet, we calculated the average $z$ coordinate $z_{upper}$ of the lipid head beads in the upper leaflet in this bin, the average $z$ coordinate $z_{lower}$ of the lipid head beads in the lower leaflet, and the midplane coordinate $z_{midplane}$ of the bin as the average of $z_{upper}$ and $z_{lower}$. We finally performed a bilinear interpolation of the calculated midplane coordinates of the square lattice with Mathematica 13 to determine $z$ coordinates also for bins with less than two lipid head beads in the upper or lower leaflet, and to obtain continuous midplane shapes. We determined these membrane midplane shapes for simulations frames at intervals of 20 ns, so for example, 201 frames of a coarse-grained simulation trajectory with a length of 4 μs. The membrane midplane shape of atomistic simulation conformations was calculated similarly, for a square lattice of 17 × 17 equally sized bins because of the smaller membrane size in our atomistic simulations, using the coordinates of the common phosphorus atoms in the lipid heads for bin allocation and averaging. Along the atomistic simulation trajectories, we determined membrane midplane shapes for simulation frames at intervals of 1 ns.

## Average membrane shapes and lipid densities

To calculate average membrane midplane shapes from the continuous midplane shapes of individual conformations, we first aligned the membrane shapes by rotations around an axis in z-direction through the center of mass of the ion channel in the center of the simulation box. In this rotational alignment, the rotation angles were chosen so that the $x - y$ coordinates of the centers of mass of the three helices 38 forming the channel in the underlying simulation conformations were aligned. To generate average shapes that reflect the 3-fold symmetry of the Piezo proteins, we performed additional rotations by 120 and 240 degrees of all rotationally aligned membrane shapes and included the two additional resulting shapes per membrane shape in the averaging. In our figures and calculations, the average membrane shapes have a circular projected area with a diameter $d_m$ that corresponds to the average $x$ and $y$ dimensions of the simulation boxes of the conformations, because only these 'inscribed' circular membrane segments are overlying in all rotated conformations. In an analogous way, we calculated average, threefold symmetric lipid densities from the interpolated lipid numbers in the bins of the square lattice of each conformation.

## Excess area and bending energy

To numerically determine the excess area and bending energy of average membrane midplane shapes, we discretized the continuous average shapes and lipid densities obtained after the interpolation, rotation, and averaging steps described above by discretizing the reference $x - y$ plane of the Monge parametrization into a square lattice with lattice constant $a = 1$ nm. This lattice constant is chosen to be smaller than the bin width of about 2 nm used in determining the membrane shape of the simulation conformations, to take into account that the averaging of these membrane shapes can lead to a higher resolution compared to the 2 nm resolution of the individual membrane shapes. The excess area at site $(x, y)$ was calculated as

$$\delta A(x, y) = a^2 \left( \sqrt{1 + \left(z(x+a, y) - z(x, y)\right)^2/a^2 + \left(z(x, y+a) - z(x, y)\right)^2/a^2} - 1 \right) \tag{4}$$

where $z(x, y)$ denotes the midplane $z$ coordinate at site $(x, y)$, and the overall excess area $\Delta A$ of the average shape was obtained by summing up the excess areas $\delta A(x, y)$ at all sites of the shape, that is at all sites with $x^2 + y^2 \leq (d_m/2)^2$. To compensate for incomplete equilibration of the membrane shapes within the simulation times of the trajectories, we divide the trajectories into five equal time intervals, determine the excess area $\Delta A$ of the nanodome shape obtained from averaging over conformations of all independent simulation runs in this time interval, and linearly extrapolate the $\Delta A$ values to long timescales as shown in *Figure 4*. The $\Delta A$ values obtained from these extrapolations are plotted in *Figure 3e*. For a lattice constant of $a = 2$ nm, we obtain extrapolated values of the excess area $\Delta A$ from the coarse-grained simulations that are 2–3% lower than the values for $a = 1$ nm, which is small compared to the statistical uncertainties with relative errors of around 10%. From the atomistic

simulations of the Piezo 2 protein-membrane nanodome, we obtain extrapolated values of $\Delta A$ for a lattice constant of $a = 2$ nm that are 3–6% lower than the values for $a = 1$ nm.

We determined the bending energy at site $(x, y)$ based on the general expression for the mean curvature $H$ in Monge parametrization as

$$e_b(x, y) = 2\kappa H(x, y)^2 \mathrm{d}A(x, y) = \kappa a^2 \frac{z_{xx}\left(1 + z_y^2\right) + z_{yy}\left(1 + z_x^2\right) - 2z_{xy}z_x z_y}{2\left(1 + z_x^2 + z_y^2\right)^{5/2}} \tag{5}$$

with bending rigidity $\kappa$ and the standard discretizations of the partial derivatives

$$z_x = \left(z(x + a, y) + z(x - a, y)\right)/2a\,, \quad z_y = \left(z(x, y + a) + z(x, y - a)\right)/2a \tag{6}$$

$$z_{xx} = \left(z(x + a, y) + z(x - a, y) - 2z(x, y)\right)/a^2\,, \quad z_{yy} = \left(z(x, y + a) + z(x, y - a) - 2z(x, y)\right)/a^2 \tag{7}$$

$$z_{xy} = \left(z(x + a, y + a) + z(x - a, y - a) - z(x + a, y - a) - z(x - a, y + a)\right)/4a^2 \tag{8}$$

The overall bending energy of the average membrane midplane shape of the membrane in the protein-membrane nanodome was then calculated as

$$E_b = \sum_{x,y} e_b(x, y)\rho(x, y) \tag{9}$$

where $\rho(x, y) \leq 1$ is the relative average lipid density at site $(x, y)$, calculated as the number of lipid head beads at site $(x, y)$ divided by the mean number of lipid head beads for 'pure' membrane patches far away from the protein. To compensate for incomplete equilibration, the bending energy $E_b$ of the average membrane shapes obtained in the different time intervals of our coarse-grained simulations is extrapolated to long timescales akin to the excess area $\Delta A$ (see *Figure 4—figure supplement 1*). For a lattice constant of $a = 2$ nm, we obtain extrapolated values of $E_b$ from the coarse-grained simulations of Piezo 2 at the tension values $\gamma = 0, 1.4, 2.8, 5.5$, and 10.8 mN/m that are about 10 to 13% lower than the values for $a = 1$ nm. These lower values of $E_b$ for $a = 2$ nm likely result from the lower resolution of the membrane curvature in the averaged nanodome, rather than from remaining membrane roughness, because the averaged nanodome shapes obtained from our coarse-grained simulations are rather smooth (see *Figures 1a, 2a and b*). The averaged nanodome shapes obtained from our atomistic simulations, in contrast, exhibit remaining roughness (see *Figure 3a and b*). In contrast to the excess area $\Delta A$, the bending energy is scale invariant and, thus, rather sensitive to remaining roughness on small length scales. We therefore do not determine bending energies for the nanodome shapes obtained from our atomistic simulations.

## TM helices

In determining the average COMs of the 4-TM units in *Figures 1d and 2e*, we discarded residues of TM helix ends that protrude out of the membrane. To identify these residues for the Piezo 1 and Piezo 2 TM helices, we calculated the average contact probability of all TM helix residues with lipid tails in the final 2 μs of the 10 coarse-grained trajectories with tensionless membranes. A TM helix residue was taken to be in contact with lipid tails in a simulation frame if the smallest distance between the beads of this residue to lipid tail beads was less than 0.6 nm. We discarded consecutive stretches of residues at the helix ends with an average contact probability of less than 10% as helix ends that protrude out of the membrane.

## Acknowledgements

This research has been funded by the Deutsche Forschungsgemeinschaft (DFG) through grant CRC 1114 /A4 and by the Max Planck Society.

## Additional information

### Funding

| Funder | Grant reference number | Author |
|---|---|---|
| Deutsche Forschungsgemeinschaft | CRC 1114/A4 | Sneha Dixit |

The funders had no role in study design, data collection and interpretation, or the decision to submit the work for publication. Open access funding provided by Max Planck Society.

### Author contributions

Sneha Dixit, Data curation, Investigation, Visualization, Writing – original draft, Writing – review and editing; Frank Noé, Conceptualization, Supervision, Funding acquisition, Writing – review and editing; Thomas R Weikl, Conceptualization, Formal analysis, Supervision, Funding acquisition, Methodology, Writing – original draft, Writing – review and editing

### Author ORCIDs

Sneha Dixit ⬤ https://orcid.org/0009-0008-4874-8823
Thomas R Weikl ⬤ https://orcid.org/0000-0002-0911-5328

Reviewer #1 (Public review): https://doi.org/10.7554/eLife.105138.3.sa1
Reviewer #2 (Public review): https://doi.org/10.7554/eLife.105138.3.sa2
Reviewer #3 (Public review): https://doi.org/10.7554/eLife.105138.3.sa3
Author response https://doi.org/10.7554/eLife.105138.3.sa4

## Additional files

### Supplementary files

MDAR checklist

### Data availability

The molecular dynamics data of this article are available in the Edmond data repository at https://doi.org/10.17617/3.USULVG.

The following dataset was generated:

| Author(s) | Year | Dataset title | Dataset URL | Database and Identifier |
|---|---|---|---|---|
| Dixit S, Weikl TR | 2025 | Coase-grained and atomistic simulation trajectories of membrane-embedded Piezo 1 and 2 proteins | https://doi.org/10.17617/3.USULVG | Edmond, 10.17617/3.USULVG |

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
