## [Editor Report · eLife Assessment]

This work represents an **important** contribution to our understanding of how membrane energetics influence protein conformation and function in mechano-sensitive channels. Through extensive molecular dynamics simulations and energetic analysis, the study **convincingly** demonstrates how the channel structure is shaped by a balance of protein and membrane-induced forces, effectively reconciling experimental data from different membrane environments. This work will appeal broadly to researchers and readers with interests in ion channel structure and function, mechanosensation, and membrane biophysics.

---

## [Referee Report · Reviewer #1 (Public review)]

Dixit, Noe, and Weikl apply coarse-grained and all-atom molecular dynamics to determine the response of the mechanosensitive proteins Piezo 1 and Piezo 2 proteins to tension. Cryo-EM structures in micelles show a high curvature of the protein whereas structures in lipid bilayers show lower curvature. Is the zero-stress state of the protein closer to the micelle structure or the bilayer structure? Moreover, while the tension sensitivity of channel function can be inferred from experiment, molecular details are not clearly available. How much does the protein's height and effective area change in response to tension? With these in hand, a quantitative model of its function follows that can be related to the properties of the membrane and the effect of external forces.

Simulations indicate that in a bilayer the protein relaxes from the highly curved cryo-EM dome (Figure 1).

Under applied tension the dome flattens (Figure 2) including the underlying lipid bilayer. The shape of the system is a combination of the membrane mechanical and protein conformational energies (Eq. 1). The membrane mechanical energy is well-characterized. It requires only the curvature and bending modulus as inputs. They determine membrane curvature and the local area metric (Eq. 4) by averaging the height on a grid and computing second derivatives (Eqs. 7, 8) consistent with known differential geometric formulas.

While I am still critical generally of a precise estimate of the energy from simulated membrane shapes (after all it is not trivial to precisely determine even the bending modulus from a simulation), I believe with their revision the authors have convinced me that their estimate is a high quality one, without obvious issues. Although there appears to have been a miscommunication about increasing the density of grain or lowering the density of grain, the authors have tried two grains and determined a similar deformation energy, which addresses my concern. Furthermore, they have computed a dramatically reduced simplification of the curve and determined a similar value.

In summary, this paper uses molecular dynamics simulations to quantify the force of the Piezo 1 and Piezo 2 proteins on a lipid bilayer using simulations under controlled tension, observing the membrane deformation, and using that data to infer protein mechanics. While much of the physical mechanism was previously known, the study itself is a valuable quantification.

---

## [Referee Report · Reviewer #2 (Public review)]

Summary:

In this study the authors suggest that the structure of Piezo2 in a tensionless simulation is flatter compared to the electron microscopy structure. This is an interesting observation and highlights the fact that the membrane environment is important for Piezo2 curvature. Additionally, the authors calculate the excess area of Piezo2 and Piezo1, suggesting that it is significantly smaller compared the area calculated using the EM structure or simulations with restrained Piezo2. Finally, the authors propose an elastic model for Piezo proteins. Those are very important findings, which would be of interest to the mechanobiology field.

Whilst I like the suggestion that the membrane environment will change Piezo2 flatness, could this be happening because of the lower resolution of the MARTINI simulations? In other words, would it be possible that MARTINI is not able to model such curvature due to its lower resolution?

Related to my comment above, the authors say that they only restrained the secondary structure using an elastic network model. Whilst I understand why they did this, Piezo proteins are relatively large. How can the authors know that this type of elastic network model restrains, combined with the fact that MARTINI simulations are perhaps not very accurate in predicting protein conformations, can accurately represent the changes that happen within Piezo channel during membrane tension?

Modelling or Piezo1, seems to be based on homology to Piezo2. However, the authors need to further evaluate their model, e.g. how it compares with an Alphafold model.

To calculate the tension induce flattening of Piezo channel, the authors "divide all simulation trajectories into 5 equal intervals and determine the nanodome shape in each interval by averaging over the conformations of all independent simulation runs in this interval.". However, probably the change in the flattening of Piezo channel happens very quickly during the simulations, possibly within the same interval. Is this the case? and if yes does this affects their calculations?

Finally, the authors use a specific lipid composition, which is asymmetric. Is it possible that the asymmetry of the membrane causes some of the changes in the curvature that they observe? Perhaps more controls, e.g. with a symmetric POPC bilayer is needed to identify whether membrane asymmetry plays a role in the membrane curvature they observe.

---

## [Referee Report · Reviewer #3 (Public review)]

Strengths:

This work focuses on a problem of deep significance: quantifying the structure-tension relationship and underlying mechanism for the mechanosensitive Piezo 1 and 2 channels. Such an objective is challenging for molecular dynamics simulations, due to the relatively large size of each membrane-protein system. Nonetheless, the approach chosen here is based on methodology that is, in principle, established and widely accessible. Therefore, another group of practitioners would likely be able to reproduce these findings with reasonable effort.

More specifically, while acknowledging the limitations of the MARTINI force field, this work makes a significant improvement compared to previous simulations of Piezo proteins by adopting a range of membrane tensions that includes physiologically relevant values (below 10 mN/m).

Weaknesses:

The two main results of this paper are (1) that both channels exhibit a flatter structure compared to cryo-EM measurements, and (2) their estimated force vs. displacement relationship. Although the former correlates at least quantitatively with prior experimental work, the latter relies exclusively on simulation results and model parameters.

My remaining technical concerns in the revised manuscript are as follows:

(1) At each membrane tension, all concurrent atomistic simulations were initialized from the same snapshot of a previous CG simulation: in my opinion, it is inaccurate to refer to those atomistic simulations as "independent" from each other (as is done twice in the caption of Figure 3, as well as in the text).

(2) Continuum mechanics calculations were employed to model the membrane's curvature energetics. The bending modulus, k, was not determined for the specific lipid composition used in this study, but was instead taken from previous MARTINI simulations involving the same primary lipid, POPC. Given that these calculations are intended to describe MARTINI simulations specifically, this approximation may be acceptable. However, it does not account for the increased stiffness observed in POPC/cholesterol mixtures-an effect measured experimentally but not reproduced by the MARTINI model-nor does it reflect the asymmetric conditions, as all referenced simulations involve symmetric bilayers. As a result, the bending energies and forces shown in Figure 5(c,d) are internally consistent within the model, but they probably correspond to real values up to an unknown multiplicative factor.

---

## [Author Response]

The following is the authors’ response to the original reviews.

**Reviewer #1 (Public review):**
Dixit, Noe, and Weikl apply coarse-grained and all-atom molecular dynamics to determine the response of the mechanosensitive proteins Piezo 1 and Piezo 2 proteins to tension. Cryo-EM structures in micelles show a high curvature of the protein whereas structures in lipid bilayers show lower curvature. Is the zero-stress state of the protein closer to the micelle structure or the bilayer structure? Moreover, while the tension sensitivity of channel function can be inferred from the experiment, molecular details are not clearly available. How much does the protein's height and effective area change in response to tension? With these in hand, a quantitative model of its function follows that can be related to the properties of the membrane and the effect of external forces.Simulations indicate that in a bilayer the protein relaxes from the highly curved cryo-EM dome (Figure 1).Under applied tension, the dome flattens (Figure 2) including the underlying lipid bilayer. The shape of the system is a combination of the membrane mechanical and protein conformational energies (Equation 1). The membrane's mechanical energy is well-characterized. It requires only the curvature and bending modulus as inputs. They determine membrane curvature and the local area metric (Equation 4) by averaging the height on a grid and computing second derivatives (Equations 7, 8) consistent with known differential geometric formulas.The bending energy can be limited to the nano dome but this implies that the noise in the membrane energy is significant. Where there is noise outside the dome there is noise inside the dome. At the least, they could characterize the noisy energy due to inadequate averaging of membrane shape.My concern for this paper is that they are significantly overestimating the membrane deformation energy based on their numerical scheme, which in turn leads to a much stiffer model of the protein itself.

We agree that “thermal noise” is intrinsic to MD simulations, as in “real” systems, leading to thermally excited shape fluctuations of membranes and conformational fluctuations of proteins. However, for our coarse-grained simulations, the thermally excited membrane shape fluctuations can be averaged out quite well, and the resulting average shapes are smooth, see e.g. the shapes and lines of the contour plots in Fig. 1 and 2. For our atomistic simulations, the averaged shapes are not as smooth, see Fig. 3a and the lines of the contour plots in Fig. 3b. Therefore, we do not report bending energies for the nanodome shapes determined from atomistic simulations, because bending energy calculations are sensitive to remaining “noise” on small scales (due to the scale invariance of the bending energy), in contrast to calculations of excess areas, which we state now on lines 620ff.

For our coarse-grained simulations, we now corroborate our bending energy calculations based on averaged 3d shapes by comparing to bending energy values obtained from highly smoothened 2d mean curvature profiles (see Fig. 1c for mean curvature profiles in tensionless membranes). We discuss this in detail from line 323 on, starting with:

“To corroborate our bending energy calculations for these averaged three-dimensional nanodome shapes, we note that essentially identical bending energies can be obtained from the highly smoothened mean curvatures *M* of the two-dimensional membrane profiles. …”

Two things would address this:(1) Report the membrane energy under different graining schemes (e.g., report schemes up to double the discretization grain).

There are two graining schemes in the modeling, and we have followed the reviewer’s recommendation regarding the second scheme. In the first, more central graining scheme, we use quadratic membrane patches with a sidelength of about 2 nm to determine membrane midplane shapes and lipid densities of each simulation conformation. This graining scheme has also been previously employed in Hu, Lipowsky, Weikl, PNAS 38, 15283 (2013) to determine the shape and thermal roughness of coarse-grained membranes. A sidelength of 2 nm is necessary to have sufficiently many lipid headgroups in the upper and lower leaflet in the membrane patches for estimating the local height of these leaflets, and the local membrane midplane height as average of these leaflet heights (see subsection “Membrane shape of simulation conformation” in the Methods section for details). However, we strongly believe that doubling the sidelength of membrane patches in this discretization is not an option, because a discretization length of 4 nm is too coarse to resolve the membrane deformations in the nanodome, see e.g. the profiles in Fig. 1b. Moreover, any “noise” from this discretization is rather completely smoothened out in the averaging process used in the analysis of the membrane shapes, at least for the coarse-grained simulations. This averaging process requires rotations of membrane conformations to align the protein orientations of the conformations (see subsection “Average membrane shapes and lipid densities” for details). Because of these rotations, the original discretization is “lost” in the averaging, and a continuous membrane shape is generated. To calculate the excess areas and bending energies for this smooth, continuous membrane shape, we use a discretization of the Monge plane into a square lattice with lattice parameter 1 nm. As a response to the referee’s suggestion, we now report that the results for the excess area do not change significantly when doubling this lattice parameter to 2 nm. On line 597, we write:

“For a lattice constant of a=2 nm, we obtain extrapolated values of the excess area Delta A from the coarse-grained simulations that are 2 to 3% lower than the values for a=1 nm, which is a small compared to statistical uncertainties with relative errors of around 10%.”

On lines 614ff, we now state that the bending energy results are about 10% to 13% lower for a=2 nm, likely because of the lower resolution of the curvature in the nanodome compared to a=1 nm, rather than incomplete averaging and remaining roughness of the coarse-grained nanodome shapes.

(2) For a Gaussian bump with sigma=6 nm I obtained a bending energy of 0.6 kappa, so certainly in the ballpark with what they are reporting but significantly lower (compared to 2 kappa, Figure 5 lower left). It would be simpler to use the Gaussian approximation to their curves in Figure 3 - and I would argue more accurate, especially since they have not reported the variation of the membrane energy with respect to the discretization size and so I cannot judge the dependence of the energy on discretization. I view reporting the variation of the membrane energy with respect to discretization as being essential for the analysis if their goal is to provide a quantitative estimate for the force of Piezo. The Helfrich energy computed from an analytical model with a membrane shape closely resembling the simulated shapes would be very helpful. According to my intuition, finite-difference estimates of curvatures will tend to be overestimates of the true membrane deformation energy because white noise tends to lead to high curvature at short-length scales, which is strongly penalized by the bending energy.

Instead of Gaussian bumps, we now calculate the membrane bending energy also from the two-dimensional, continuous mean curvature profiles (see Fig. 1c). These mean curvature profiles are highly smoothened (see figure caption for details). Nonetheless, we obtain essentially the same bending energies as in our discrete calculations of averaged, smoothened threedimensional membrane shapes, see new text on lines 326ff. We believe that this agreement corroborates our bending energy calculations. We still focus on values obtained for threedimensional membrane shapes, because of incomplete rotational symmetry. The three-dimensional membrane shapes exhibit variations with the three-fold symmetry of the Piezo proteins, see Figure 2a and b.

We agree that the bending energy of thermally rough membranes depends on the discretization scheme, because the discretization length of any discretization scheme leads to a cut-off length for fluctuation modes in a Fourier analysis. But again, we average out the thermal noise, for reasons given in the Results section, and analyse smooth membrane shapes.

The fitting of the system deformation to the inverse time appears to be incredibly ad hoc ... Nor is it clear that the quantified model will be substantially changed without extrapolation. The authors should either justify the extrapolation more clearly (sorry if I missed it!) or also report the unextrapolated numbers alongside the extrapolated ones.

We report the values of the excess area and bending energy in the different time intervals of our analysis as data points in Fig. 4 with supplement. We find it important to report the time dependence of these quantities, because the intended equilibration of the membrane shapes in our simulations is not “complete” within a certain time window of the simulations. So, just “cutting” the first 20 and 50% of the simulation trajectories, and analysing the remaining parts as “equilibrated” does not seem to be a reasonable choice here, at least for the membrane properties, i.e. for the excess area and bending energy. We agree that the linear extrapolation used in our analysis is a matter of choice. At least for the coarse-grained simulations, the extrapolated values of excess areas and bending energies are rather close to the values obtained in the last time windows (see Figure 4).

In summary, this paper uses molecular dynamics simulations to quantify the force of the Piezo 1 and Piezo 2 proteins on a lipid bilayer using simulations under controlled tension, observing the membrane deformation, and using that data to infer protein mechanics. While much of the physical mechanism was previously known, the study itself is a valuable quantification. I identified one issue in the membrane deformation energy analysis that has large quantitative repercussions for the extracted model.
**Reviewer #2 (Public review):**
Summary:In this study, the authors suggest that the structure of Piezo2 in a tensionless simulation is flatter compared to the electron microscopy structure. This is an interesting observation and highlights the fact that the membrane environment is important for Piezo2 curvature. Additionally, the authors calculate the excess area of Piezo2 and Piezo1, suggesting that it is significantly smaller compared to the area calculated using the EM structure or simulations with restrained Piezo2. Finally, the authors propose an elastic model for Piezo proteins. Those are very important findings, which would be of interest to the mechanobiology field.Whilst I like the suggestion that the membrane environment will change Piezo2 flatness, could this be happening because of the lower resolution of the MARTINI simulations? In other words, would it be possible that MARTINI is not able to model such curvature due to its lower resolution?Related to my comment above, the authors say that they only restrained the secondary structure using an elastic network model. Whilst I understand why they did this, Piezo proteins are relatively large. How can the authors know that this type of elastic network model restrains, combined with the fact that MARTINI simulations are perhaps not very accurate in predicting protein conformations, can accurately represent the changes that happen within the Piezo channel during membrane tension?

These questions regarding the reliability of the Martini model are very reasonable and are the reason why we include also results from atomistic simulations, at least for Piezo 2, and compare the results. In the Martini model, secondary structure constraints are standard. In addition, constraints on the tertiary structure (e.g. via an elastic network model) are also typically used in simulations of soluble, globular proteins. However, such tertiary constraints would make it impossible to simulate the tension-induced flattening of the Piezo proteins. So instead, as we write on lines 427ff, “we relied on the capabilities of the Martini coarse-grained force field for modeling membrane systems with TM helix assemblies (Sharma and Juffer, 2013; Chavent et al., 2014; Majumder and Straub, 2021).” In these refences, Martini simulations were used to study the assembly of transmembrane helices, leading to agreement with experimentally observed structures. As we state in our article, our atomistic simulations corroborate the Martini simulations, with the caveats that are now more extensively discussed in the new last paragraph of the Discussion section starting on line 362.

Modelling or Piezo1, seems to be based on homology to Piezo2. However, the authors need to further evaluate their model, e.g. how it compares with an Alphafold model.

We understand the question, but see it beyond the scope of our article, also because of the computational demand of the simulations. The question is: Do coarse-grained simulations of Piezo1 based on an Alphafold model as starting structure lead to different results? It is important to note that we only model the rather flexible 12 TM helices at the outer ends of the Piezo 1 monomers via homology modeling to the Piezo 2 structure, which includes these TM helices. For the inner 26 TM helices, including the channel, we use the high-quality cryo-EM structure of Piezo 1. Alphafold may be an alternative for modeling the outer 12 helices, but we don’t think this would lead to statistically significant differences in simulations – e.g. because of the observed overall agreement of membrane shapes in all our Piezo 1 and Piezo 2 simulation systems.

To calculate the tension-induced flattening of the Piezo channel, the authors "divide all simulation trajectories into 5 equal intervals and determine the nanodome shape in each interval by averaging over the conformations of all independent simulation runs in this interval.". However, probably the change in the flattening of Piezo channel happens very quickly during the simulations, possibly within the same interval. Is this the case? and if yes does this affect their calculations?

Unfortunately, the flattening is not sufficiently quick, so is not complete within the first time windows, see data points in Figure 4. We therefore report the time dependence with the plots in Figure 4 and extrapolate, see also our response above to reviewer 1.

Finally, the authors use a specific lipid composition, which is asymmetric. Is it possible that the asymmetry of the membrane causes some of the changes in the curvature that they observe? Perhaps more controls, e.g. with a symmetric POPC bilayer are needed to identify whether membrane asymmetry plays a role in the membrane curvature they observe.

Because of the rather high computational demands, such controls are beyond our scope. We don’t expect statistically significant differences for symmetric POPC/cholesterol bilayers. On lines 229ff, we now state:

“Our modelling assumes that any spontaneous curvature from asymmetries in the lipid composition is small compared to the curvature of the nanodome and, thus, negligible, which is plausible for the rather slight lipid asymmetry of our simulated membranes (see Methods).”

**Reviewer #3 (Public review):**
Strengths:This work focuses on a problem of deep significance: quantifying the structure-tension relationship and underlying mechanism for the mechanosensitive Piezo 1 and 2 channels. This objective presents a few technical challenges for molecular dynamics simulations, due to the relatively large size of each membrane-protein system. Nonetheless, the technical approach chosen is based on the methodology that is, in principle, established and widely accessible. Therefore, another group of practitioners would likely be able to reproduce these findings with reasonable effort.Weaknesses:The two main results of this paper are (1) that both channels exhibit a flatter structure compared to cryo-EM measurements, and (2) their estimated force vs. displacement relationship. Although the former correlates at least quantitatively with prior experimental work, the latter relies exclusively on simulation results and model parameters.Below is a summary of the key points we recommend addressing in a revised version of the manuscript:(1) The authors should report and discuss controls for the membrane energy calculations, specifically by increasing the density of the discretization graining. We also suggest validating the bending modulus used in the energy calculations for the specific lipid mixture employed in the study.

We have addressed both points, see our response to the reviewer’s comments for further details.

(2) The authors should consider and discuss the potential limitations of the coarse-grained simulation force field and clarify how atomistic simulations validate the reported results, with a more detailed explanation of the potential interdependencies between the two.

We now discuss the caveats in the comparison of coarse-grained and atomistic simulations in more detail in a new paragraph starting on line 362.

(3) The authors should provide further clarification on other points raised in the reviewers' comments, for instance, the potential role of membrane asymmetry.

We have done this – see above. We now further explain on lines 437ff why we use an asymmetric membrane. On lines 230ff, we discuss that any spontaneous membrane curvature due to lipid asymmetry is likely small compared to the nanodome curvature and, thus, negligible.

**Reviewer #1 (Recommendations for the authors):**
(1) Report discretization dependence of the membrane energy (up to double the density of the current discretization graining).

We have added several text pieces in the paragraph “Excess area and bending energy” starting on line 583 in which we state how the results depend on the lattice constant a of the calculations.

(2) Evaluate an analytical energy of a membrane bump with a shape similar to the simulation. This would be free of all sampling and discretization artifacts and would thus be an excellent lower bound of the energy.

We have done this for the curvature profile in Figure 1c and corresponding curvature profiles of the shape profiles in Figure 2d, see next text on lines 326ff.

Minor:(1) The lipid density (Figure 1 right, 2c, 3c) is not interesting nor is it referred to. It can be dropped.

We think the lipid density maps are important for two reasons: First, they show the protein shape obtained after averaging conformations, as low-lipid-density regions. Second, the lipid densities are used in the calculation of the bending energies, to limit the bending energy calculations to the membrane in the nanodome, see Eq. 9. We therefore prefer to keep them.

(2) Figure 7 is attractive but not used in a meaningful way. I suggest inserting the protein graphic from Figure 7 into Figure 1 with the 4-helix bundles numbered alongside the structure. Figure 7 could then be dropped.

Figure 7 is a figure of the Methods section. We need it to illustrate and explain aspects of the setup (numbering of helices, missing loops) and analysis (numbering scheme of 4-TM helix units).

(3) Some editing of the use of the English language would be helpful. "Exemplary" is a bit of a funny word choice, it implies that the conformation is excellent, and not simply representative. I'd suggest "Representative conformation".

We agree and have replaced “exemplary” by “representative”.

(4) Typos:Equation 4 - Missing parentheses before squared operator inside the square root.

We have corrected this mistake.

**Reviewer #2 (Recommendations for the authors):**
This study focuses mainly on Piezo2; the authors do not perform any atomistic simulations of Piezo1, and the coarse-grained simulations for Piezo1 are shorter. As a result, their analysis for Piezo2 seems more complete. It would be good if the authors did similar studies with Piezo1 as with Piezo2.

We agree that atomistic simulations of Piezo 1 would be interesting, too. However, because the atomistic simulations are particularly demanding, this is beyond our scope.

**Reviewer #3 (Recommendations for the authors):**
(1) At line 63, a very large tension from the previous work by De Vecchis et al is reported (68 mN/m). The authors are sampling values up to about 21 mN/m, which is considerably smaller. However, these values greatly exceed what typical lipid membranes can sustain (about 10 mN/m) before rupturing. When mentioning these large tensions, the authors should emphasize that these values are not physiologically significant, because they would rupture most plasma membranes. That said, their use in simulation could be justified to magnify the structural changes compared to experiments.

We agree that our largest membrane tension values are unphysiological. However, we see a main novelty and relevance of our simulations in the fact that we obtain a response of the nanodome in the physiological range of membrane tensions, see e.g. the 3^rd^ sentence of the abstract. Yes, we include simulations at tensions of 21 mN/m, but most of our simulated tension values are in the range from 0 to 10 mN/m (see e.g. Fig. 3e), in contrast to previous simulation studies.

(2) At line 78 and in the Methods, only the reference paper is for the CHARMM protein force field, but not for the lipid force field.

We have added the reference Klauda et al., 2010 for the CHARMM36 lipid force field in both spots.

(3) (Line 83) Acknowledging that the authors needed to use the structure from micelles (because it has atomic resolution), how closely do their relaxed Piezo structures compare with the lowerresolution data from the MacKinnon and Patapoutian papers?

There are no structures reported in these papers to compare with, only a clear flattening as stated.

(4) (Line 99) The authors chose a slightly asymmetric lipid membrane composition to capture some specific plasma-membrane features. However, they do not discuss which features are described by this particular composition, which doesn't include different acyl-chain unsaturations between leaflets. Further, they do not seem to comment on whether there is enrichment of certain lipid species coupled to curvature, or whether there is any "scrambling" occurring when the dome section and the planar membrane are stitched together in the preparation phase (Figure 8).

Enrichment of lipids in contact with the protein is addressed in the reference Buyan et al., 2020, based on Martini simulations with Piezo 1. We have a different focus, but still wanted to keep an asymmetric membrane as in essentially all previous simulation studies as now stated also on lines 439ff, to mimic the native Piezo membrane environment. There is no apparent “scrambling” in the setup of our membrane systems. We also did not explore any coupling between curvature and lipid composition, but will publish the simulation trajectories to enable such studies.

(5) (Caption of Figure 2). Please comment briefly in the text why the tensionless simulation required a longer simulation run (e.g. larger fluctuations?)

We added as explanation on line 500 as explanation: “ … to explore the role of the long-range shape fluctuations in tensionless membranes for the relaxation into equilibrium”. The relaxation time of membrane shape fluctuations strongly increases with the wave length, which is only limited by the simulation box size in the absence of tensions. However, also for 8 microsecond trajectories, we do not observe complete equilibriation and therefore decided to extrapolate the excess area and bending energy values obtained for different time intervals of the trajectories.

(6) (Caption of Figure 3). Please clarify in the Methods how the atomistic simulations were initialized were they taken from independent CG simulation snapshots? If not, the use of the adjective "independent" would be questionable given the very short atomistic simulation time length.

We now added that the production simulations started from the same structure. On lines 386, we now discuss the starting structure of the atomistic simulations in more detail.

(7) (Line 202). The approach of discretizing the bilayer shape is reasonable, but no justification was provided for the 1-nm grid spacing. In my opinion, there should be a supporting figure showing how the bending energy varies with the grid spacing.

We now report also the effect of a 2-nm grid spacing on the results, see new text passages on page 18, and provide an explanation for the smaller 1-nm grid spacing on lines 587ff, where we write:

“This lattice constant [a = 1 nm] is chosen to be smaller than the bin width of about 2nm used in determining the membrane shape of the simulation conformations, to take into account that the averaging of these membrane shapes can lead to a higher resolution compared to the 2 nm resolution of the individual membrane shapes.”

(8) (Line 211). The choice by the authors to use a mixed lipid composition complicates the task of defining a reasonable bending modulus. Experimentally and in atomistic simulations, lipids with one saturated tail (like POPC or SOPC) are much stiffer when they are mixed with cholesterol (https://doi.org/10.1529/biophysj.105.067652, https://doi.org/10.1103/PhysRevE.80.021931, https://doi.org/10.1093/pnasnexus/pgad269). On the other hand, MARTINI seems to predict a slight *softening* for POPC mixed with cholesterol (https://doi.org/10.1038/s41467-023-43892-x). Further complicating this matter, mixtures of phospholipids with different preferred curvatures are predicted to be softer than pure bilayers (e.g. https://doi.org/10.1021/acs.jpcb.3c08117), but asymmetric bilayers are stiffer than symmetric ones in some circumstances (https://doi.org/10.1016/j.bpj.2019.11.3398).This issue can be quite thorny: therefore, my recommendation would be to either: (a) directly compute k for their lipid composition, which is straightforward when using large CG bilayers (as was done in Fowler et al, 2016), but it would also require more advanced methods for the atomistic ones; (b) use a reasonable *experimental* value for k, based on a similar enough lipid composition.

We now justify in somewhat more detail why we use an asymmetric membrane, but agree that his complicates the bending energy estimates. We only aim to estimate the bending energy in the Martini 2.2 force field, because our elasticity model is based on and, thus, limited to results obtained with this force field. We have included the two further references using the Martini 2.2 force field suggested by the reviewer on line 213, and discuss now in more detail how the bending rigidity estimate enters and affects the modeling, see lines 226ff.

(9) (Line 224). Does this closing statement imply that all experimental work from ex-vivo samples describe Piezo states under some small but measurable tension?

We compare here to the cryo-EM structure in detergent micelles. So, there is no membrane tension, there may be a surface tension of the micelle, but we assume here that Piezo proteins are essentially force free in detergent micelles. Membrane embedding, in contrast, leads to strong forces on Piezo proteins already in the absence of membrane tension, because of the membrane bending energy.

(10) (Line 304). The Discussion concludes with a reasonable point, albeit on a down note: could the authors elaborate on what kind of experimental approach may be able to verify their modeling results?

Very good question, but this is somewhat beyond our expertise. We don’t have a clear recommendation – it is complicated. What can be verified is the flattening, i.e. the height and curvature of the nanodome in lower-resolution experiments. We see our results in line with these experiments, see Introduction.

(11) (Line 331). The very title of the Majumder and Straub paper addresses the problem of excessive binding strength between protein beads in the MARTINI force field, which should be mentioned. Figure 3(d) shows that the atomistic systems have larger excess areas than the CG ones. This could be related to MARTINI's "stickiness", or just statistical sampling. Characterizing the grid spacing (see point 7 above) might help illuminate this.

We discuss now the larger excess area values of the atomistic simulations on lines 381ff.

(12) (Lines 367, 375). Are the harmonic restraints absolute position restraints or additional bonds?

Note also that the schedule at which the restraints are released (10-ns intervals) is relatively quick. Does the membrane have enough time to equilibrate the number of lipids in each leaflet?

These are standard, absolute position restraints. The 10-ns intervals may be too short to fully equilibrate the numbers of lipids, we have not explored this. The main point in the setup was to have a reasonable TM helix embedding with a smooth membrane, without any rupturing. This turned out to be tricky, with the procedures illustrated in Figure 8 as solution. If the membrane is smooth, the lipid numbers quickly equilibrate either in the final relaxation or in the initial nanoseconds of the production runs.

(13) (Line 387) The use of an isotropic barostat for equilibration further impedes the system's ability to relax its structure. I feel that the authors should validate more strongly their protocol to rule out the possibility that incomplete equilibration could bias dynamics towards flatter membranes, which is one of the main results of this paper.

We don’t see how choices in the initial relaxation steps could have affected our results, at least for the coarse-grained simulations. There is more and more flattening throughout all simulation trajectories, see e.g. the extrapolations in Figure 4. All initial simulation structures are significantly less flattened than the final structures in the production runs.

(14) (Line 403). What is the protocol for reducing the membrane size for atomistic simulation? This is even more important to mention than for CG simulations.

We just cut lipids beyond the intended box size of the atomistic simulations. As a technical point, we now have also added on line 507 how PIP2 lipids were converted.

(15) (Line 423). The CHARMM force field requires a cut-off distance of 12 Å for van der Waals forces, with a force-based continuous switching scheme. The authors should briefly comment on this deviation and its possible impact on membrane properties. Quick test simulations of very small atomistic bilayers with the chosen composition could be used as a comparison.

We don’t expect any relevant effect on membrane properties within the statistical accuracies of the quantities of interest here (i.e. excess areas).

(16) (Equation 4). There are some mismatched parentheses: please check.

We have corrected this mistake.

(17) (Equations 7-8). Why did the authors use finite-differences derivatives of z(x,y) instead of using cubic splines and the corresponding analytical derivatives?

In our experience, second derivatives of standard cubic splines can be problematic. The continuous membrane shapes we obtain in our analysis are averages of such splines. We find standard finite differences more reliable, and therefore discretize these shapes. Already for the 2d membrane profiles of Figure 1b and 2d, calculating curvatures from interpolations using splines is problematic.